

# Unraveling the Impact of Heterogeneity and Morphology on Light Absorption Enhancement of Black Carbon-Containing Particles

**Jing Wei[1], Jin-Mei Ding[2], Yao Song[1], Xiao-Yuan Wang[2], Xiang-Yu Pei[1], Sheng-Chen Xu[2], Fei Zhang[1], Zheng-Ning Xu[1], Xu-Dong Tian[2], Bing-Ye Xu[2], Zhi-Bin Wang[1,3]**

[1] State Key Laboratory of Soil Pollution Control and Safety, Zhejiang Provincial Key Laboratory of Organic Pollution Process and Control, College of Environmental and Resource Sciences, Zhejiang University, Hangzhou 310058, China

[2] Ecological and Environmental Monitoring Center of Zhejiang Province, Hangzhou 310012, China

[3] ZJU-Hangzhou Global Scientific and Technological Innovation Center, Zhejiang University, Hangzhou 311200, China

**Correspondence:** Bing-Ye Xu (xubingye@zjemc.org.cn) and Zhi-Bin Wang (wangzhibin@zju.edu.cn)





**Abstract**

Black carbon (BC) is a strong climate forcer, but considerable uncertainty remains in
estimating its radiative impact, largely due to persistent gaps between observed and
modeled light absorption enhancement ($E_{abs}$). In this study, we employed a Centrifugal
Particle Mass Analyzer and Single Particle Soot Photometer tandem system to
characterize mass ratio ($M_R$, coating-to-BC) and morphology of BC-containing
particles in Hangzhou, China. Fortunately, low, medium, and high $E_{abs}$ values were
observed during a single field campaign. Results show that the uniform core-shell Mie
model overestimated $E_{abs}$ especially in clean conditions (low $E_{abs}$). A morphology-
dependent correction scheme was developed to improve optical property estimates of
BC in the "transition state." This improved model better reproduces measured $E_{abs}$ in
different pollution conditions and reveals that the concentrations of particle chemical
composition affect the $M_R$ threshold defining this state. Our findings highlight the need
to account for real-world particle complexity in climate-relevant BC modeling.

**1 Introduce**


Black carbon (BC) is a strongly light-absorbing aerosol that effectively absorbs
solar radiation, warms the atmosphere, and contributes to direct radiative forcing
(DRF)(Bond and Bergstrom, 2006; Seinfeld, 2008). According to IPCC assessments,
the global effective DRF of BC ranges from −0.28 to 0.41 W/m$^2$ (Szopa et al., 2021).
Most climate models(Bauer et al., 2013; Chen et al., 2024; Stier et al., 2005; Wang et
al., 2023; Zhang et al., 2025) use Mie theory to estimate BC light absorption
enhancement ($E_{abs}$), assuming a uniform core-shell structure where BC core is fully
coated by coating materials. This approach predicts a monotonic increase in $E_{abs}$ with
the coating-to-core mass ratio ($M_R$), often reaching values up to ~2, consistent with
laboratory results (Peng et al., 2016). However, field observations commonly report
lower $E_{abs}$ values, typically around 1.4 and sometimes as low as 1.09 (Cappa et al., 2012;
Huang et al., 2024). This discrepancy mainly stems from the oversimplified
assumptions in Mie theory, which fail to capture the real atmospheric complexity in BC
size distribution, morphology, and mixing state (Wang et al., 2021c). These limitations
introduce considerable uncertainty in assessing the radiative impacts of BC. Therefore,
improving our understanding of the mechanisms controlling BC $E_{abs}$ and its
atmospheric evolution is essential for reducing uncertainties in BC DRF estimates and
enhancing the accuracy of climate model projections.
Previous studies have explored the discrepancies in BC $E_{abs}$ from various
perspectives. A recent particle-resolved model study reveals that particle-to-particle
heterogeneity in $M_R$ significantly influences $E_{abs}$ (Fierce et al., 2020). Traditional
spherical core-shell models tend to overestimate $E_{abs}$, especially for partially coated
particles with low $M_R$. However, $M_R$ heterogeneity alone cannot explain low $E_{abs}$ under
high $M_R$ conditions (Huang et al., 2024). Atmospheric BC particles also vary in
morphology. Fresh BC exhibit a branched structure that collapses into compact shapes
with reduced light absorption cross-sections during aging (Moteki and Kondo, 2007;
Romshoo et al., 2024; Li et al., 2024). Early aging stage feature uneven coatings, while
aged particles show BC core either encapsulated or located near the particle surface.



These configurations further influence $E_{abs}$. Recent studies suggest that the proportion
of non-spherical BC particles and the position of the BC core may be key factors
contributing to low $E_{abs}$, leading to an overestimation by core-shell model (Huang et al.,
2024; Chen et al., 2024; Zhang et al., 2022). Thus, due to $M_R$ heterogeneity and the
morphological complexity, the mechanisms driving BC $E_{abs}$ remain poorly understood,
necessitating further research into relevant physical and chemical processes.

65         In this study, a suite of state-of-the-art instruments were employed to
simultaneously capture the magnitude and dynamic evolution of $M_R$ and morphology
of BC-containing particles in Hangzhou, China. Fortunately, field measurements
directly revealed the coexistence of high, medium, and low $E_{abs}$ under high bulk-
averaged $M_R$ conditions. Further, the influences of $M_R$ and morphology of BC-
containing particles on $E_{abs}$ are quantified to reconcile the discrepancies between
models and observations under different $E_{abs}$ conditions. Subsequently, an improved
model for "transition-state" BC-containing particles was developed to reproduce the
observed $E_{abs}$ under varying $E_{abs}$ and bulk-averaged $M_R$ conditions. This study
underscores the importance of simultaneously considering $M_R$ heterogeneity and
morphology of BC-containing particles in predicting $E_{abs}$. These findings contribute to
reducing the uncertainty in estimating the DRF of BC.

## 2 Methods

### 2.1 Overview of the field campaign and instrumentation

The field measurements were conducted at the Central Air Quality Assurance
Monitoring Station (30.25°N,120.24°E) in Hangzhou from 3th Sept., 2023 to 13th Oct.,
2023. The sampling site is located just 100 meters from the Qiantang River in the
western part of Hangzhou, with major traffic routes within 3 kilometers to the northeast
and southwest of the station. The schematic of the instrumentation is provided in Fig.
S1. Aerosols were sampled after passing through a $PM_{2.5}$ impactor and then dried
through a diffusion dryer before reaching subsequent instruments. The mass of an BC-
containing particle ($M_p$) and of the BC core ($M_{BC}$) can be simultaneously obtained by a
Centrifugal Particle Mass Analyzer (CPMA, Cambustion) and a single-particle soot
photometer (SP2, DMT Inc.) tandem system. In the CPMA-SP2 system, particles with
known mass ($M_p$) selected by CPMA were injected into the SP2, and the $M_p$ was set
from 0.9 fg to 30 fg with a logarithmic evenly spaced distribution, divided into 10 $M_p$,
specially 0.93 fg, 1.37 fg, 2.02 fg, 2.97 fg, 4.36 fg, 6.40 fg, 9.39 fg, 13.78 fg, 20.22 fg
and 29.68 fg, then the mass of BC core was measure by SP2. The duration of one set
point cycle was 1 hour, with each $M_p$ point sampling for 5 minutes, and all $M_p$ points
was sampling for total of 50 minutes. The remaining 10 minutes were divided into 4
minutes for instrument stabilization and 6 minutes for measuring all BC-containing
particles when the valve was switched to the single SP2 line. During further data
analysis, particles with $M_p = 0.93$ fg and $M_p = 1.37$ fg exhibited excessively noisy
scattering signals and were therefore excluded from subsequent statistical analysis. The
SP2 was calibrated with size-resolved Aquadag aerosols to establish the correlation



between incandescence peak height and the mass of BC-containing particles (DMT,
2011) (Fig. S2c and d). As reported in previous studies (Liu et al., 2020; Liu et al., 2014;
Zhang et al., 2018), a correction factor of 0.75 was applied to the peak height during
calibration. The SP2 scattering signal was calibrated with polystyrene latex spheres
(PSL) of known sizes (210 nm, 270 nm and 310 nm) (Fig. S2b). Additionally, the
calibration of the scattering and the incandescence channels was performed before and
after the measurement campaign. The mass concentrations of non-refractive OA, nitrate,
sulfate, ammonium and chloride was measured by an aerosol chemical speciation
monitor (ACSM, Aerodyne Research Inc.).
The aerosol extinction coefficient and scattering (Fig. S3) at wavelength of 440
nm, 530 nm and 630 nm was measured by Multi-Wavelength Cavity Attenuated Phase
Shift Single-Scattering Albedo Monitor (CAPS-ALB) (Weber et al., 2022), the aerosol
absorption coefficient was the difference between extinction coefficient and scattering
coefficient. Meanwhile, the aerosol scattering coefficient at wavelength of 450 nm, 525
nm, and 635 nm was also measured by Nephelometer (Multi Wavelength Integrating
Nephelometer) (Schloesser, 2016). The slope of the scattering coefficient measured by
CAPS and Nephelometer at corresponding wavelength was close to 1 (Fig. S4),
indicating the all data are reliable for further analysis. Besides, before sampling, the
scattering coefficient of CAPS and Nephelometer at every wavelength was calibrated
using PSL spheres. The slope of the scattering coefficient measured by CAPS (or the
Nephelometer) and modeled by Mie theory was close to 1 (Fig. S5), indicating the
reliability of the CAPS and Nephelometer. The lower detection limit of the
Nephelometer at all three wavelengths was 0.3 Mm$^{-1}$ with a 60-second integration time,
while that of the CAPS was 1 Mm$^{-1}$ with 30-second integration time.

## 2.2 Mixing state and morphology of the particle-resolved BC-containing particles

The mixing state of single BC-containing particle can be represented by the mass
ratio of the BC coating to BC core without any assumptions,

$$M_R = \left(M_p - M_{BC}\right)/M_{BC} \tag{1}$$

where $M_p$ and $M_{BC}$ were the mass and core of each BC-containing particle. Then $M_R$
was converted to the bulk-averaged $M_R$ to compared the measured $E_{abs}$ in bulk particles
by summing of total coating and core mass of BC-containing particles every hour,

$$bulk\ average\ M_R = \frac{\sum_i M_{R,i} \times M_{BC,i}}{\sum_i M_{BC,i}} \tag{2}$$

where $i$ was the $i^{th}$ single BC-containing particle. The data measured by CPMA-SP2
were corrected via several steps, including (1) correction of delay time, (2) multi-
charged particles and (3) collection efficiency (for details, see Text S1).
In CPMA-SP2, when knowing $M_p$ and $M_{BC}$, the modeled scattering cross section
($C_{modeled}$) of BC-containing particles can be derived using Mie theory (Wang et al.,
2021a). This calculation assumes a core-shell structure with BC having refractive index
of 2.26–1.26$i$ (Liu et al., 2017; Zhao et al., 2020) and the coating with 1.48$i$, along with
a coating density of 1.5 g cm$^{-3}$ (Liu et al., 2015). The measured scattering cross section
($C_{measured}$) is analyzed using the leading-edge-only (LEO) technique, which reconstructs



the distorted scattering signal when BC-containing particles passes through the SP2
laser beam, as widely described in previous studies (Liu et al., 2014; Zhang et al., 2016;
Brooks et al., 2019; Gao et al., 2007; Zhang et al., 2020). Note only particles with
successfully fitted LEO signals are considered in the optical property calculations.
Subsequently, $C_{modeled}$ can be compared with $C_{measured}$ to infer the morphological
characteristic of BC-containing particles (Liu et al., 2017; Liu et al., 2020). Further
details can be seen in Section 3.
**2.3 The measured and modeled $E_{abs}$**
The light absorption enhancement of BC-containing particles is defined as the ratio
of the mass absorption cross section (MAC) of the coated and uncoated BC-containing
particles,

$$E_{abs\_measured} = \frac{MAC_{BC}}{MAC_{BC\_core}} \tag{3}$$

where $E_{abs\_measured}$ is the measured light absorption enhancement, and $MAC_{BC\_core}$ is the
mass absorption coefficient for uncoated BC particles. The value of $MAC_{BC\_core}$ was
obtained by extrapolating $MAC_{BC}$ to the limit of bulk-averaged $M_R = 0$. The $MAC_{BC\_core}$
at wavelength of 630 nm was 9.08 (Fig. S6).
The commonly used models for calculating the optical properties of BC-
containing particles include Core-shell Mie theory (Cappa et al., 2012), T-matrix (Wu
et al., 2020), and discrete dipole approximation (DDA) (Kahnert and Kanngießer, 2020).
Among them, T-matrix, and DDA fully account for the impact of morphology of BC-
containing particles by incorporating detailed three-dimensional parameters such as
fractal dimension and monomer number (Wu et al., 2020). In contrast, Core-shell Mie
theory relies solely on the BC core size ($D_c$) and coating thickness ($D_p/D_c$). Given the
measurement data available in this study, the Core-shell Mie theory was used to
calculate the $E_{abs}$ of BC-containing particles in this study. The refractive index ($RI$) of
BC and its coatings are assumed to be n=1.85+0.71$i$ and n=1.5+0$i$ (Liu et al., 2015; Liu
et al., 2014). The size of BC core and coating thickness was directly measured by
CPMA-SP2 tandem system. For the uniform core-shell assumption, the $M_R$ of every $D_c$
was equal to bulk-averaged $M_R$ (Method), as described by Cappa et al. (2019) and Liu
et al. (2017). Then the particle-resolved Core-shell Mie theory was employed to
calculate the $MAC_{BC}$ of individual BC-containing particles. After obtaining the
particle-resolved MAC, we can calculate the MAC of BC particle ensembles as

$$MAC_{BC} = \sum_{i=0}^{n} MAC_i \times \frac{[BC]_i}{[BC]} \tag{4}$$

where $MAC_i$ and $[BC]_i$ are the particle-resolved $MAC_{BC}$ and mass concentration of the
BC core, [BC] is the mass concentration of BC particle ensembles, and $n$ is the number
of $M_p$ bins. Then, the $E_{abs}$ of BC particle ensembles are calculated as the ratio of $MAC_{BC}$
to $MAC_{BC\_core}$. Note that the $MAC_{BC\_core}$ here is calculated using the core-shell Mie
model when $D_p / D_c = 1$.
The morphology-dependent approach was based on the $C_{measured}$ measured by SP2
and $C_{modeled}$ by Core-shell Mie theory at wavelength of 1064 nm. The approach mainly
primarily focuses on calculating the optical properties of transition-state BC-containing





particles based on their morphology. In this study, we use the scattering cross-section
of transition-state BC-containing particles measured by SP2 at 1064 nm as a reference,
establish a mathematical relationship between the $C_{measured}$ and $M_R$ (Fig. 3d, e and f),
and use this relationship to calculate the MAC of this portion of BC-containing particles.
Then the bulk-averaged MAC was obtained by integrating the MAC of each BC-
containing particle.

**3 Results and discussions**

**3.1 Direct observation of different $E_{abs}$.**

The average $E_{abs\_measured}$ during the sampling period in Hangzhou is 1.28, and the bulk-
averaged $M_R$ is 3.32. Based on the measured $E_{abs}$, three scenarios were identified (Fig.
S7 and Fig. 1). Case 1 spans from September 3 to September 23, 2023, and from
October 1 to October 7, 2023. Case 2 covers the period from September 24 to
September 30, 2023, while Case 3 runs from October 8 to October 13, 2023. $E_{abs\_measured}$
in Case 1 is relatively low, with an average value of 1.09. In contrast, the $E_{abs\_measured}$ in
Case 2 and Case 3 is significantly higher, with average values of 1.84 and 1.55,
respectively.

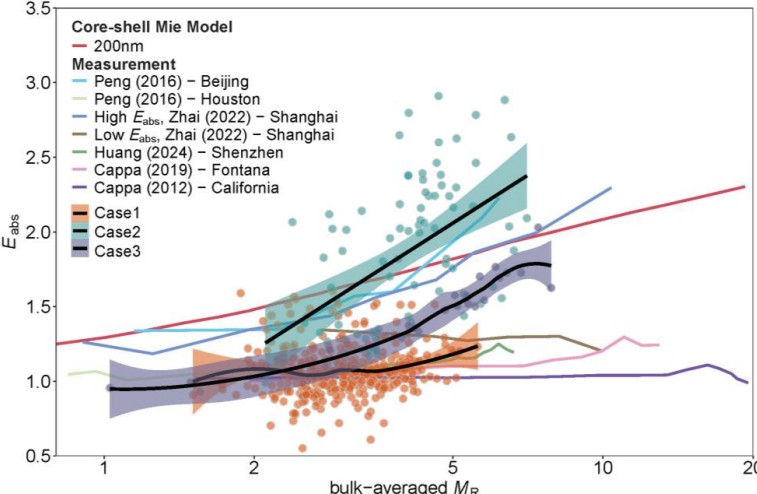

**Figure. 1** Comparison of measured $E_{abs}$ in different observation periods and with
previous studies (Peng et al., 2016; Cappa et al., 2012; Zhai et al., 2022; Cappa et al.,
2019; Huang et al., 2024). The black solid line represents the fitted smoothing curve of
bulk-averaged $M_R$ and measured $E_{abs}$, with the shaded area indicating the 95%
confidence interval of the fit.

Some studies conducted in cities such as Beijing (Peng et al., 2016), Shanghai
(Zhai et al., 2022) have observed a notable increase in $E_{abs\_measured}$. However, in cleaner
regions like Shenzhen (Huang et al., 2024), Houston (Peng et al., 2016) and California



(Cappa et al., 2012; Cappa et al., 2019), even when $R_{BC}$ reached approximately 5, many
studies reported minimal increases in $E_{abs\_measured}$. Fortunately, our observations
captured a wide range of $E_{abs\_measured}$ values (0.92~1.84), encompassing high, medium,
and low levels. In these three cases, $E_{abs\_measured}$ showed markedly different evolution
patterns during the aging of BC-containing particle populations (Fig. 1). Specially, in
Case 1, $E_{abs\_measured}$ remained almost unchanged with increasing bulk-averaged $M_R$. In
Case 3, $E_{abs\_measured}$ showed a significant increase as bulk-averaged $M_R$ rose, while in
Case 2, the increase in $E_{abs\_measured}$ was even more pronounced for the same increment
in bulk-averaged $M_R$. Besides, in Case 2, the $E_{abs}$ calculated by the traditional core-shell
Mie model ($E_{abs\_uniform}$) closely matches the $E_{abs\_measured}$, followed by Case 3, which
shows a slightly lower level of consistency. However, in Case 1, the $E_{abs\_uniform}$ predicted
by the traditional core-shell Mie model is significantly higher than the $E_{abs\_measured}$. In
our subsequent analysis, we will address this discrepancy by exploring both
heterogeneity of $M_R$ and morphology.

**3.2 Role of mixing state heterogeneity in $E_{abs}$ and direct evidence of morphology**
**of BC-containing particles with increasing coating thickness.**

The measurement data were integrated and corrected to quantify the coating-to-
black carbon mass ratio ($M_R$) and morphology of each black carbon-containing particle.
The $M_R$ is an important indicator reflecting the aging of BC (Zeng et al., 2024; Li et al.,
2024; Liu et al., 2017). Fig. 2a and 2b illustrated significant differences in the
normalized number distribution of BC-containing particles at $M_p$ = 4.35 fg and 9.38 fg
at different period. Specifically, during Case 2 and Case 3, the $M_R$ presents a unimodal
distribution, with the peak value increasing with increasing $M_p$. In contrast, during Case
1, the $M_R$ exhibits a distinct bimodal distribution, and both peak positions shift toward
higher $M_R$ values as $M_p$ increases. For example, when $M_p$ = 4.35 fg, the two peaks occur
at $M_R$ = 1 fg and 4.2 fg, respectively, whereas at $M_p$ = 9.38 fg, they shift to $M_R$ = 1.8 fg
and 8.0 fg, respectively. The standard deviation ($SD$) of $\log_{10}(M_R)$ was used to
characterize the heterogeneity of $M_R$ among individual BC-containing at each $M_p$. The
results showed that the $SD$ of Case 1 (0.63) was greater than that of Case 3 ($SD$ = 0.52),
followed by Case 2 ($SD$ = 0.48). In contrast, the $E_{abs\_measured}$ exhibited an opposite trend
during three corresponding periods, indicating that the greater $M_R$ heterogeneity of BC-
containing particles leads to a lower $E_{abs\_measured}$. This is further supported by the
positive correlation between the difference in $E_{abs}$ (modeled by the uniform core-shell
Mie model and measured) and $SD$ (Fig. 2d). In other words, the uniform core-shell
model predicts a greater $E_{abs}$ as $M_R$ heterogeneity increases, which leads to a greater
discrepancy between the measured and modeled $E_{abs}$. This discrepancy is primarily due
to stems from the uniform core-shell model's simplified treatment of $M_R$ heterogeneity
in BC (Romshoo et al., 2024; Wang et al., 2021c).



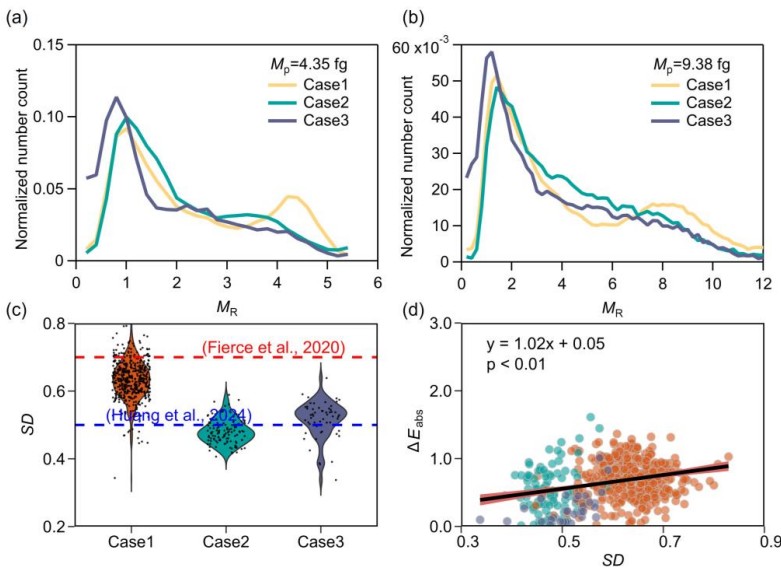


**Figure. 2** Heterogeneity of BC-containing particles in different observation periods.
The average $M_R$ normalized number distribution of BC-containing particles at $M_p =$
4.35 fg (**a**) and $M_p = 9.38$ fg (**b**) during different cases. (**c**) Comparison of the average
standard deviation ($SD$) of $\log_{10}(M_R)$ in this study with those reported by Huang et al.
(2024) and Fierce et al. (2020), the $SD$ characterizes the dispersion of $M_R$ among
individual BC-containing particles. (**d**) Linear fitting of $\Delta E_{abs}$ ($E_{abs\_uniform}$-$E_{abs\_measured}$)
with the $SD$ of $\log_{10}(M_R)$.

The SP2 can measure the scattering cross-section ($C_{sca}$) of single BC-containing
particles, providing insights into their morphology. Fig. 3 presents the variation of the
ratio $C_{sca\_measured}/C_{sca\_modeled}$ with $M_R$ under different $M_p$. When $M_R$ is relatively low,
$C_{sca\_measured}/C_{sca\_modeled}$ is less than 1, suggesting that the BC core may exist in a fractal
structure, remain bare, or are not fully embedded in the coating materials. Consequently,
the measured $C_{sca}$ is lower than that calculated by the core-shell Mie model. This
observation aligns with the finding by Liu et al. (2017), who classified such BC-
containing particles as externally mixed. As $M_R$ increases, the ratio of
$C_{sca\_measured}/C_{sca\_modeled}$ also increases, indicating the BC core becomes more compact
and more thoroughly coated, transitioning toward a core-shell structure. Previous
studies (Liu et al., 2017; Liu et al., 2020) have referred to BC in this aging stage as
being in the "transition state". However, the $M_R$ criteria for the "transition state" vary
across different periods: 1.43-3.78 in Case 2, 1.45-4.19 in Case 3, and higher range of
1.78-6.34 in Case1. This implies that under polluted conditions (Case 2 and Case 3),
BC particles can form core-shell structure with less coating material. This can be
attributed, on the one hand, to unfavorable meteorological conditions—particularly low
wind speed (0.81 m/s)—which facilitated the accumulation and formation of secondary
inorganics and organics, accelerating the heterogeneous aging of BC-containing
particles. On the other hand, the presence of abundant pollutants originating from





Jiangsu and northern Zhejiang (Fig. S7 and Fig. S8) further contributed to this rapid
aging process. These factors explain why BC required less coating material to evolve
into a core-shell structure in Case 2 and Case 3 compared to Case 1. When $M_R$ exceeds
the range associated with the transition state, BC-containing particles predominantly
adopt a spherical shape, as further confirmed by the steady ratio of
$C_{sca\_measured}/C_{sca\_modeled}$.

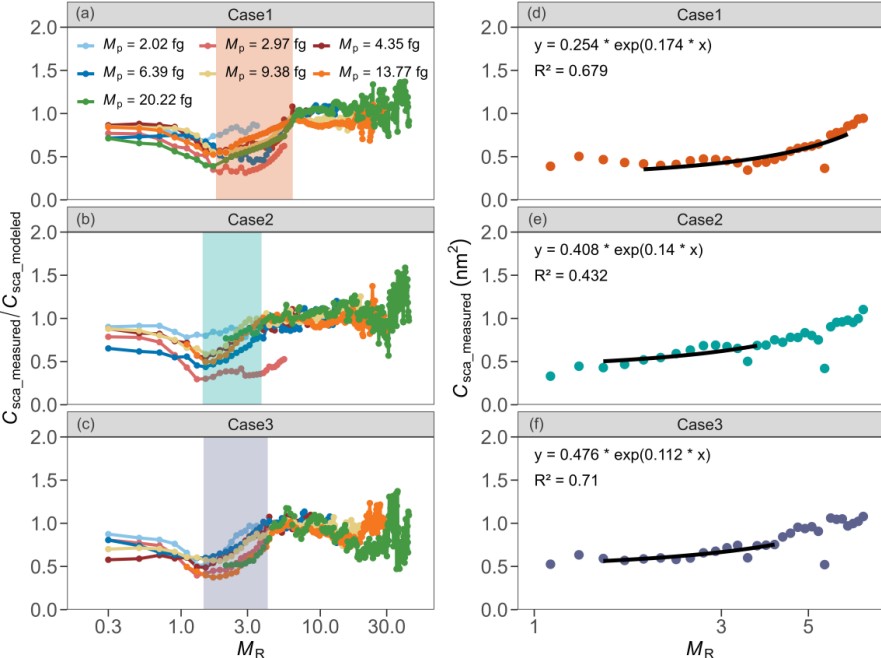


**Figure. 3** The morphology and measured optical properties of BC-containing particles
as a function of mass ratio ($M_R$). (**a**) – (**c**) show the ratio of measured to modeled
scattering cross-section during different cases. The shadows indicate the $M_R$ ranges
corresponding to "transition-state" BC-containing particles. (**d**) – (**f**) present the
measured scattering cross-section as a function of $M_R$, along with fitted morphology-
dependent models representing the "transition-state" BC-containing particles.

### 3.3 The predicted $E_{abs}$ during different period.

The complexity of BC in the atmosphere depends on various factors, including the
size, morphology of the BC core, coating amount, and the interaction between the BC
core and the coating. this study retrieves comprehensive multidimensional information
on single BC-containing particle, which is subsequently incorporated into the optical
model, as shown in Fig. 4. The bias between $E_{abs\_uniform}$ and $E_{abs\_measured}$ varies across
different periods, even when applying the same model input scheme. Specially, the
$E_{abs\_uniform}$ closely aligns with $E_{abs\_measured}$ during Case 2 and Case 3, with deviations
below 10%, whereas the deviation increases to as much as to 65% during Case 1,
primarily due to the higher $M_R$ dispersion of BC-containing particles in this period. To
further investigate this discrepancy, we assume that all BC-containing particles adopt a



core-shell morphology and calculate the $E_{abs}$ of each BC-containing particle based on
the measured single-particle $M_R$. Subsequently, the $E_{abs}$ of bulk BC-containing particles
was determined (Method) and compared with $E_{abs\_measured}$ to evaluate their consistency.
The results show that for the Case 1, although the discrepancy between the measured
and modeled values exhibits a decreasing trend, the average deviation remains as high
as 38%. This finding is inconsistent with previous studies (Fierce et al., 2020), which
suggest that the $M_R$ heterogeneity of BC-containing particles can largely explain the
overestimation of $E_{abs\_measured}$ by the uniform core-shell model. The discrepancy arises
because the $M_R$ dispersion of BC-containing particles in that study was lower than the
observed value (Fig. 2c) in the present study. However, for Case 2 and Case 3 with
higher $M_R$ heterogeneity of BC-containing particles, the error between the model and
measured $E_{abs}$ is almost negligible, with deviations below 10%, indicating that the core-
shell Mie model can reproduce the observed $E_{abs}$ during these periods. These findings
further validate that the degree of $M_R$ dispersion of BC-containing particles is a key
factor in determining whether the core-shell Mie model overestimates the observed $E_{abs}$,
and to what extent this overestimation occurs.

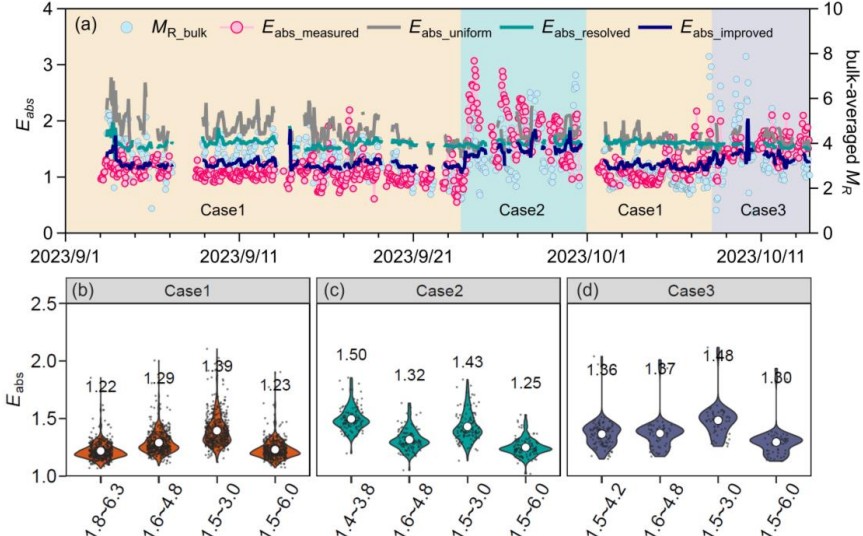


**Figure. 4** Comparison of measured and modeled $E_{abs}$ under various different model
input schemes. (**a**) shows the time series of the $E_{abs}$ during the observation period. (**b**)
– (**d**) were the sensitivity of $E_{abs}$ to the "transition state" range of BC-containing
particles. The "transition state" range of 1.6~4.8 was the average value derived from
Case 1, Case 2 and Case 3. The range of 1.5~3.0 and 1.5~6.0 was reported by Liu et al.
(2017), and Liu et al. (2020), respectively.
The proportion of BC cores embedded in their coatings varies significantly due to
differences in atmospheric conditions, which further leads to differences in $E_{abs}$. Under
low non-volatile $PM_1$ and low relative humidity (RH) conditions, the proportion of BC
cores embedded in the coating was small, resulting in lower $E_{abs}$ (Cappa et al., 2019).
However, during the haze period in Beijing (Zeng et al., 2024), intense liquid-phase



chemical reactions produced a large number of inorganic substances, causing rapid
aging of BC-containing particles, which significantly increased the degree of BC core
embedding and led to higher $E_{abs}$. The essence of this differences lies in the fact that the
BC core lacks sufficient coating material to form a core-shell structure. Therefore, the
embedding pattern mainly occurs in the BC-containing particles with small $M_R$. The
chemical composition of $PM_1$ especially for nitrate during Case 2 and Case 3 was
significantly higher than in Case 1 (Fig. S7), resulting in higher bulk-averaged $M_R$ and
$E_{abs}$ for BC. However, during Case 1, the lower concentrations of chemical components
of $PM_1$ was unfavorable for the formation of BC coatings, thereby leading to lower
bulk-averaged $M_R$ and $E_{abs}$. Therefore, a larger proportion of BC-containing particles
remain in a "transition" and externally mixed state (Fig. S9), for which the $E_{abs}$ is
considered negligible. Consequently, accurately quantifying the $E_{abs}$ of "transition state"
is essential for determine the bulk $E_{abs}$. To address this, a morphology-dependent
approach was developed to calculate $E_{abs}$ of BC-containing particles in the "transition
state", which was achieved by fitting the measured $C_{sca}$ against $M_R$ (Fig. 3d-3f). The
improved model yielded an average $E_{abs}$ of only 1.21 during Case 1. For Case 2 and
Case 3, although the modeled $E_{abs}$ ($E_{abs\_improved}$) were slightly lower than $E_{abs\_measured}$, the
deviation remained within 20%. These findings suggest that the improved model
effectively reconcile the $E_{abs}$ of BC-containing particles under different atmospheric
conditions.

343          In recent years, the application of particle-resolved models in multiple field
observations has effectively improved the overestimation of observed $E_{abs}$ by uniform
core-shell Mie model (Fierce et al., 2020; Li et al., 2024; Jiang et al., 2025). Some
studies have also considered the morphology of individual BC-containing particle and
developed advanced models such as the electron-microscope-to-BC-simulation (EMBS)
(Wang et al., 2021c; Wang et al., 2021b; Chakrabarty et al., 2006), thereby enhancing
the accuracy of modeled BC $E_{abs}$. Although the improved model proposed in this study
demonstrates better fitting performance especially under low $PM_1$ concentrations
conditions (clean periods, Case 1), its accuracy remains highly dependent on the
classification of the "transition state." During heavily polluted periods or in highly
polluted regions (Case 2 and Case 3), the "transition state" is usually concentrated
within a smaller $M_R$ range, whereas under clean periods, it usually appears in a larger
$M_R$ range. If a uniform classification of "transition state" is applied, it will inevitably
introduce errors in modeled BC $E_{abs}$ (Fig. 4b, 4c and 4c). Moreover, considering only
the $M_R$ heterogeneity of BC-containing particle can effectively reproduce the observed
BC $E_{abs}$ in polluted environments. However, applying different model input schemes to
varying atmospheric conditions would increase the complexity of radiation model.
Therefore, a unified model input scheme that incorporates both $M_R$ heterogeneity and
morphology is recommended for all atmospheric conditions. This model input scheme
allows for a more consistent representation of BC mixing state while focusing solely
on how environmental variations influence the classification of the "transition state".
In summary, improvements to the model should not only account for the $M_R$
heterogeneity, morphological differences of BC-containing particle but also incorporate
the dynamic changes in environmental conditions to more accurately predict the optical



properties of BC-containing particles in different atmospheric environments, thereby reducing errors in assessing the climate effects of BC.

**4 Conclusions**

In this study, we employed the CPMA-SP2 tandem system to investigate the mass ratio of coating and core ($M_R$), and morphology of BC-containing particles in Hangzhou, China, and to assess how heterogeneity in $M_R$ and morphology influence their $E_{abs}$ under different atmospheric conditions. By dividing the observation into three representative scenarios (Case 1, Case 2, and Case 3), we observed significant differences in the measured $E_{abs}$, that were closely associated with the evolution and distribution of $M_R$ and morphology characteristics. The findings indicate that both $M_R$ heterogeneity and morphology play critical roles in modeling $E_{abs}$. During clean condition (Case 1), the uniform core-shell Mie model significantly overestimated $E_{abs}$, while during polluted periods (Case 2 and Case 3), model predictions were more consistent with the measured $E_{abs}$. To address these discrepancies, we developed a morphology-dependent correction scheme for the optical property calculation of BC in the "transition state" by incorporating $M_R$ heterogeneity and morphology information. The improved model input method effectively reconciled the measured $E_{abs}$ across varying pollution conditions, especially in clean environments dominated by externally mixed and partially coated BC-containing particles. Our results further show that environmental factors-such as the chemical composition of $PM_1$-not only influence the morphology and $M_R$ distribution of BC-containing particles but also shift the threshold $M_R$ range defining the "transition state". These findings underscore the limitations of uniform model input schemes under complex atmospheric conditions and demonstrate the value of a unified modeling framework that incorporates both $M_R$ heterogeneity and particle morphology. Such improvement enables more accurate representation of BC mixing state and light absorption across diverse environmental settings, thereby reducing uncertainties in assessing the radiative forcing of BC-containing particles.

**Data availability.** The data are available from the link: http://doi.org/10.6084/m9.figshare.29097263.

**Author contributions.**

Conceptualization: Jing Wei, Zhi-Bin Wang
Data curation: Jing Wei, Yao Song, Xiang-Yu Pei, Fei Zhang, Zheng-Ning Xu, Jin-Mei Ding, Xiao-Yuan Wang, Sheng-Chen Xu, Xu-Dong Tian, Bing-Ye Xu
Formal analysis: Jing Wei
Funding acquisition: Zhi-Bin Wang, Bing-Ye Xu
Investigation: Zhi-Bin Wang
Methodology: Jing Wei, Zhi-Bin Wang
Visualization: Jing Wei, Zhi-Bin Wang
Writing – original draft: Jing Wei



**Competing interests.** At least one of the (co-)authors is a member of the editorial board of *Atmospheric Chemistry and Physics*.

**Financial support.** This research was supported by the Joint Fund of Zhejiang Provincial Natural Science Foundation of China (LZJMZ25D050003), the National Key Research and Development Program of China (2022YFC3703505), the National Natural Science Foundation of China (42305098), the China Postdoctoral Science Foundation (2023M733028), the Key Laboratory of Formation and Prevention of Urban Air Pollution Complex, Ministry of Ecology and Environment (2025080166).

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
