# Peer review of "Unraveling the Impact of Heterogeneity and Morphology on Light"

_EGUsphere, 2025_

## Author Comment (AC1)

**Response to Reviewer #1:**

The paper presents an interesting analysis of measured absorption enhancement by black carbon particles mixed with other species. The data are interesting and quite valuable. However, several aspects of the manuscript are not well explained, and some are confusing or unclear. I think the authors can address most of these shortfalls, and if so, then the manuscript would deserve publication.

General comments

**[Comment 1]** The definition of "transition state" is vague, and it should be quantified. The criteria used are not clear.

**Response:** We appreciate the reviewer's insightful comment. In the revised manuscript, we have clarified the definition and usage of the term "transition state". We now explicitly describe it as a case dependent range in which BC particles becomes more compact and thoroughly coated, gradually transitioning toward a core-shell structure. This corresponding text has been revised as follows (Lines 355-363):

"*As $M_R$ increases, $C_{sca\_measured}/C_{sca\_modeled}$ also increases, indicating the BC particles becomes more compact and more thoroughly coated, transitioning toward a core-shell structure (Corbin et al., 2023). Following previous studies (Liu et al., 2017a; Liu et al., 2020), we describe this stage as a "transition state". In this work, the transition state is neither defined by a fixed $M_R$ threshold nor by any directly observed morphological boundary. Instead, it reflects an optically inferred state in which scattering enhancement increases markedly, with $M_R$ ranges of 1.78-6.34 (Case 1), 1.43-3.78 (Case 2), and 1.45-4.19 (Case 3).*"

This clarification provides a more operational and quantitative definition consistent with the observed variability among different cases.

**[Comment 2]** How the three cases (1,2, and 3) were defined is not clear.

**Response:** We appreciate the reviewer's valuable comment. In the revised manuscript, we have clarified the criteria to define the three cases (Lines 265-282). Specifically, the classification was based on the temporal variation of the measured $E_{abs}$, with $E_{abs} = 1.5$ selected as the reference threshold, as described below:

"*To investigate the temporal evolution of BC coating, the observation period was classified into three cases based on the variation of $E_{abs\_measured}$, using $E_{abs}=1.5$ as the reference threshold (Fig.1). Case 1 (September 3-23 and October 1-7, 2023) corresponds to $E_{abs\_measured}$ significantly below 1.5. These periods were characterized by relatively low non-refractory $PM_1$ concentrations and high wind speeds (WS = 0.94 ± 0.04 m/s). Back-trajectory analysis further shows that Case 1 was dominated by clean marine and nearby local air masses, resulting in relatively clean and weakly aged conditions (Fig. 1a). Case 2 (September 24–30) corresponds to periods when $E_{abs\_measured}$ remained continuously higher than 1.5. This episode occurred under stagnant meteorological conditions-characterized by weak winds (WS = 0.81 ± 0.02 m/s) and elevated relative humidity (RH = 81.34 ± 17.12%) -that favored secondary aerosol formation. Back-trajectory analysis further indicates that Case 2 was dominated by air masses transported from Jiangsu and passing through*"

*northern Zhejiang, enhancing pollutant accumulation and promoting more aged BC conditions. Case 3 includes periods when $E_{abs\_measured}$ persistently fluctuated around 1.5. The air masses during this period were a mixture of polluted inland outflow and clean marine inflow, suggesting the air masses were moderately aged–which explains the intermediate $E_{abs}$."*

**[Comment 3]** The link with local conditions, pollution level, and contribution from different sources is not well developed. It is probably possible and relatively straightforward for the authors to use existing monitoring data to corroborate some of their discussions. I would also suggest generating some more discussion of the back-trajectory simulations to determine transport path and time, and potential sources. The authors do show some of this work in the SI, but in the main paper, there is little to no mention, and more discussion should be added.

**Response:** We thank the reviewer for this insightful comment. We agree that linking our observations with local pollution conditions and potential source influences can strengthen the interpretation of BC absorption enhancement. Although source attribution is not the primary focus of this study, we have incorporated additional analyses to address the reviewer's concerns.

First, we added a discussion connecting the observed $E_{abs}$ variations with concurrent aerosol pollution levels and meteorological conditions (Lines 264-281). These additions help contextualize the measured $M_R$ and $E_{abs}$ under different air-quality regimes. Second, in response to the reviewer's suggestion, we expanded the discussion of air-mass transport pathways in the revised manuscript. Specifically, we performed supplementary 48-hour back-trajectory simulations at 100 m, 500 m, and 1000 m above ground level (Fig. 1c and Fig. S7), and the results consistently show similar transport pathways for different observation periods. These findings indicate that the polluted air masses during Case 2 passed through northern Zhejiang, supporting the interpretation that regional-scale transport contributed to the elevated aerosol loading and high $M_R$ and $E_{abs}$ conditions. A brief discussion of these results has now been included in Section 3.3 (Lines 439-448).

Overall, these additions help strengthen the link between pollution level, transport conditions, and the observed optical properties of BC while maintaining the central focus of this work.

Lines 264-281: *"To investigate the temporal evolution of BC coating, the observation period was classified into three cases based on the variation of $E_{abs\_measured}$, using $E_{abs}$=1.5 as the reference threshold (Fig.1). Case 1 (September 3-23 and October 1-7, 2023) corresponds to $E_{abs\_measured}$ significantly below 1.5. These periods were characterized by relatively low non-refractory $PM_1$ concentrations and high wind speeds (WS = 0.94 ± 0.04 m/s). Back-trajectory analysis further shows that Case 1 was dominated by clean marine and nearby local air masses, resulting in relatively clean and weakly aged conditions (Fig. 1a). Case 2 (September 24–30) corresponds to periods when $E_{abs\_measured}$ remained continuously higher than 1.5. This episode occurred under stagnant meteorological conditions-characterized by weak winds (WS = 0.81 ± 0.02 m/s) and elevated relative humidity (RH = 81.34 ± 17.12%) -that favored secondary aerosol formation. Back-trajectory analysis further indicates that Case 2 was dominated by air masses transported from Jiangsu and passing through northern Zhejiang, enhancing pollutant accumulation and promoting more aged BC conditions. Case 3 includes periods when $E_{abs\_measured}$ persistently fluctuated around 1.5. The air masses during this period were a mixture of polluted inland outflow and clean marine inflow, suggesting the air masses were moderately aged–which explains the intermediate $E_{abs}$."*

Lines 433-443: "*During Case 2, the high aerosol loading and elevated bulk-averaged $M_R$ were largely influenced by regional transport, as air masses at 100 m, 500 m, and 1000 m all followed similar pathways from the northern Yangtze River Delta into northern Zhejiang (Fig. 1c and Fig. S7). Stagnant meteorological conditions, elevated relative humidity, and enhanced oxidative capacity further facilitated vigorous liquid-phase and photochemical reactions, promoting the abundant formation of secondary coatings on BC surfaces (Peng et al., 2016). Notably, our observations show that $E_{abs}$ increases systematically with the increasing contribution of secondary nitrate (Fig. S8), consistent with the fact that nitrate-rich conditions enhance aqueous-phase oxidation and accelerate the formation of thick inorganic coatings (Liu et al., 2017b).*"

**[Comment 4]** The SI also shows an ACSM, which might provide further information that I did not see discussed in the paper. Maybe I missed it, but was Figure S7 based on the ACSM data? Maybe that could be made clearer in the SI as well as in the main text. Small note: In the same figure, CPAS should be CAPS (?)

**Response:** We thank the reviewer for this comment. Fig. S7 has now been updated to Figure 1, , and the non-refractory components are based on the ACSM-X measurements. We have clarified this both in the revised manuscript (Lines 145-149) and the Figure 1 title (Lines 301-303) to make it explicit.

Lines 145-149: "*The mass concentrations of non-refractive OA, nitrate, sulfate, ammonium and chloride was measured by time-of-flight aerosol chemical speciation monitor with extended resolution (ToF-ACSM X, Aerodyne). Instrument principles, calibration procedures, and operational details for the ToF-ACSM X are described in a previous study of ours (Zhang et al., 2025).*"

Lines 301-303 "*The time series of BC concentrations, bulk-averaged $M_R$, measured $E_{abs}$, and the chemical components (including organics, nitrate, sulfate, ammonium and chloride) measured by TOF-ACSM X, as well as relatively humidity (RH).*"

Regarding the small note, "CPAS" in the figure has been corrected to "CAPS."

**[Comment 5]** It is not clear to me what the "improved model" is and how it is operationally defined. More details should be provided on this topic as it seems to be central to the discussion.

**Response:** We appreciate the reviewer's comment. In the revised manuscript, we have explicitly clarified what the "improved model" refers to and how it is operationally defined. The revised manuscript now provides a concise description of the observationally constrained parameterization. These details are now clearly stated in the revised manuscript (Lines 464-495).

"*In this study, we introduce an observationally constrained parameterization that links SP2-measured scattering cross sections with core-shell Mie calculations. This scheme identifies the optical transition state through following steps: (1) measuring single-particle $C_{sca}$ with SP2, (2) fitting the relationship between $C_{sca\_measured}$ and $M_R$ (Fig. 4b-4c), (3) identifying the $M_R$ range associated with transitional optical behavior ("transition state"), and (4) inferring the MAC of transition-state particles using the fitted relationship before integrating MAC over all particles to obtain bulk $E_{abs}$. This parameterization improves agreement with observations, especially during clean periods (Case 1), when the uniform core-shell assumption tends to produce the largest*

*discrepancies. However, its performance depends on correctly identifying the $M_R$ range of the transition state. Our results show that, in polluted periods (Cases 2 and 3), the $M_R$ range associated with the transition-state becomes relatively narrow, while under clean conditions it tends to expand. Consequently, applying a fixed $M_R$ threshold across all atmospheric conditions can introduce systematic biases in modeled $E_{abs}$ (Fig. 4b, c and d). Although $M_R$ heterogeneity alone can adequately reproduce $E_{abs}$ during polluted periods, adopting separate input schemes for different environments would complicate radiative transfer calculations and limit broader applicability.*

*To address this issue, we emphasize that the proposed framework is adaptable to environments in which BC particles undergo similar optical transitions. Key parameters, including the $M_R$ thresholds that define the transition state and other indicators derived from the $C_{sca\_measured}$-$M_R$ relationship, can be recalibrated for different atmospheric contexts. This includes rural aeras, biomass-burning regions, or seasons with distinct pollution characteristics, where coating composition and aging processes may vary. Although the parameterization is fundamentally based on the optical evolution of BC from loosely coated to more compact states, it can be adjusted to account for local differences in particle coating and aging dynamics. Thus, the unified scheme incorporates both $M_R$ variability and optical characteristics of transitional particles, providing a flexible and physically consistent approach for a wide range of atmospheric environments. Overall, this observationally constrained approach offers a more consistent representation of BC mixing states across diverse atmospheric conditions, thereby reducing uncertainties in optical modeling and enhancing the reliability of BC radiative effect assessments.*"

**[Comment 6]** The literature cited is somewhat lacking and perhaps a bit biased; especially lacking is the comparison with other single-particle techniques such as microscopy.

**Response:** We thank the reviewer for this valuable comment. We acknowledge that direct comparisons with single-particle microscopy observations were not included in this study. Nevertheless, we have now incorporated relevant references in the Introduction to provide additional context on the coating characteristics of ambient BC (Lines 47-59). Although microscopy techniques are not employed here, these studies highlight the variability of coating structures and the limitations of assuming idealized core–shell configurations when modeling BC absorption enhancement. This addition strengthens our discussion on the model assumptions relative to existing observational evidence and better situates our work within the framework of previous single-particle studies.

"*This discrepancy mainly stems from the oversimplified assumptions in Mie theory, which fail to capture the real atmospheric complexity in BC size distribution, coating configuration, and mixing state (Wang et al., 2021b). Previous microscopy-based single-particle studied (e.g., TEM and SEM) have visually demonstrated that the ambient BC particles exhibit diverse coating structures and highly heterogeneous mixing states, providing direct evidence of deviation from the idealized core-shell assumption (Adachi et al., 2010; Adachi and Buseck, 2013; China et al., 2013; Wang et al., 2021b). Although microscopy techniques are not employed in this work, these findings highlight the importance of realistically representing BC mixing state and coating characteristics when modeling optical properties. The mismatch between model assumptions and observations has motivated efforts to refine the conceptual modeling approaches for BC aging and coating evolution, which forms the focus of this study.*"

**[Comment 7]** Even though the paper is mostly clearly written, the grammar should still be improved; some limited examples are provided in the next section.

**Response:** We thank the reviewer for this comment. We have carefully proofread the manuscript and improved the grammar throughout the text. In addition, we have addressed the specific examples provided by the reviewer to ensure clarity and correctness of the language.

Specific comments

**[Comment 8]** Line 35: space after models

**Response: Corrected.**

**[Comment 9]** Line 51. I think Fierce's paper also discussed the effect of morphological deviations at low MR

**Response:** We thank the reviewer for this valuable suggestion. We have revised the manuscript to explicitly acknowledge that Fierce et al. (2020) discussed the effect of morphological deviations from the idealized core-shell structure, particularly at low $M_R$, on BC $E_{abs}$, described as follows (Lines 60-66):

*"A number of studies have explored the discrepancies in BC $E_{abs}$ from various perspectives. Particle-resolved modeling has demonstrated that both particle-to-particle heterogeneity in $M_R$ and deviations from the idealized core-shell structure can strongly influence absorption estimates (Fierce et al., 2020). In particular, non-uniform or partial coatings at low $M_R$ can lead to the overestimation of $E_{abs}$ by traditional core-shell models. However, these factors alone tend to cannot explain the low $E_{abs}$ frequently observed under high $M_R$ conditions (Huang et al., 2024)."*

**[Comment 10]** Line 86: "an BC" -> "a BC"

**Response: Corrected.**

**[Comment 11]** Line 111: "was" -> "were"

**Response: Corrected.**

**[Comment 12]** Line 120: How were the concentrations of PSL measured?

**Response:** We thank the reviewer for the comment. The purpose of using PSL spheres was not to measure their number concentration, but to determine their scattering cross section. Monodisperse PSL particles of different diameters were selected using a DMA and introduced into the instruments, allowing the scattering cross section to be measured for calibration purposes. This clarification has been added to the revised manuscript (Lines 164-171):

**"*Besides, before sampling, the scattering coefficient of CAPS-ALB and Nephelometer at every wavelength was calibrated using PSL spheres. Monodisperse PSL particles of different diameters (100 nm, 150 nm, 200 nm and 300 nm) were selected using a Differential Mobility Analyzer (DMA) and introduced into the instruments, enabling accurate measurement of their scattering cross section ($C_{sca}$). The slope of the $C_{sca}$ measured by CAPS-ALB (or the Nephelometer) and modeled by Mie theory was close to 1 (Fig. S5), indicating the reliability of the CAPS-ALB and Nephelometer.*"**

**[Comment 13]** Line 127: "a" in front of "single". Also, what do the authors mean by "without any assumptions"?

**Response:** We thank the reviewer for pointing this out. We have corrected the sentence to include the article "a" before "single BC-containing particle." In addition, we clarified the meaning of "without any assumptions" as follows (Lines 176-178):

*"Under the assumption of singly charged particles, the mixing state of a single BC-containing particle can be represented by the mass ratio of the BC coating to the BC core, without relying on assumptions about particle morphology or coating structure,"*

**[Comment 14]** Line 129: "to compare the" -> "to be compared with"

**Response: Corrected.**

**[Comment 15]** Line 134: Mie is good only for spherically symmetric particles. How do the authors account for that, considering the premise of the paper is the deviation from spherical symmetry?

**Response:** We thank the reviewer for this insightful comment. We fully agree that Mie theory strictly applies to spherically symmetric particles, whereas ambient BC-containing particles often deviate from this idealized assumption. In our analysis, Mie theory was used to obtain the theoretical scattering cross section ($C_{sca\_modeled}$) under the core-shell approximation. To account for discrepancies between modeled and measured $C_{sca}$, we fitted the relationship between the $C_{sca\_measured}$ and $M_R$, and applied this fit in the calculation of absorption. This approach effectively includes BC particles whose optical properties are not captured by the core-shell Mie model, including those with eccentric core-shell structures. The revised description can be found at Lines 464-473.

*"In this study, we introduce an observationally constrained parameterization that links SP2-measured scattering cross sections with core-shell Mie calculations. This scheme identifies the optical transition state through following steps: (1) measuring single-particle $C_{sca}$ with SP2, (2) fitting the relationship between $C_{sca\_measured}$ and $M_R$ (Fig. 4b-4c), (3) identifying the $M_R$ range associated with transitional optical behavior ("transition state"), and (4) inferring the MAC of transition-state particles using the fitted relationship before integrating MAC over all particles to obtain bulk $E_{abs}$. This parameterization improves agreement with observations, especially during clean periods (Case 1), when the uniform core-shell assumption tends to produce the largest discrepancies."*

**[Comment 16]** Line 137: The refractive index used here seems quite high with respect to other studies in the literature. Additionally, the coating is sometimes slightly absorbing, and that can make a difference, especially at shorter wavelengths. A sensitivity analysis using a range of indices of refraction would be helpful.

**Response:** We thank the reviewer for this comment. In this study, the refractive indices were chosen primarily to derive the $M_R$-dependent optical transition behavior of BC-containing particles from SP2 scattering measurements at 1064 nm. The BC core is assigned a complex refractive index of 2.26-1.26i, while the non-absorbing coating has a refractive index of 1.48 and a density of 1.5 g cm$^{-3}$, values that have been widely used in previous studies (Liu et al., 2017a; Zhao et al., 2020; Liu et al., 2015). Besides, since the coating in this work does not absorb at 1064 nm, its potential absorption

can be safely neglected for the purpose of deriving the $M_R$-dependent optical transition behavior. To make our purpose clearer to the readers, the main text has been revised as follows (Lines 190-209):

"*The $M_R$-dependent optical transitions of BC-containing particles were further derived from SP2 measurements at a wavelength of 1064 nm. In the CPMA-SP2 system, when both $M_p$ and $M_{BC\_core}$ are known, the modeled scattering cross section ($C_{sca\_modeled}$) of BC-containing particles can be derived using Mie theory (Wang et al., 2021a). This calculation assumes a core-shell structure, with the BC core having a refractive index of 2.26-1.26i (Liu et al., 2017a; Zhao et al., 2020) and the non-absorbing coating characterized by a refractive index of 1.48 and a density of 1.5 g cm$^{-3}$ at a wavelength of 1064 nm (Liu et al., 2015). The measured scattering cross section ($C_{sca\_measured}$) was obtained from the SP2 using the leading-edge-only (LEO) technique, which reconstructs the scattering signal as BC-containing particles pass through the SP2 laser beam due to partial evaporation of refractory-absorbing material. The validity of this reconstruction relies on the assumption that the leading-edge data used for fitting represents an unperturbed particle, as extensively reported in previous studies (Liu et al., 2014; Zhang et al., 2016; Brooks et al., 2019; Gao et al., 2007; Zhang et al., 2020). Note only particles with successfully fitted LEO signals are considered in the optical property calculations. By comparing $C_{sca\_measured}$ with $C_{sca\_modeled}$, the $M_R$-dependent optical behavior of BC-containing particles can be inferred, particularly for transition-state particles. This comparison captures how variations in coating-to-core mass ratio influence scattering, providing observational constraints on the optical evolution of BC during aging (Liu et al., 2017a; Liu et al., 2020).*"

**[Comment 17]** Line 154: MAC units?

**Response:** We thank the reviewer for pointing this out. The unit of MAC has been clarified as m$^2$ g$^{-1}$ and added to the Methods section (Line 220).

**[Comment 18]** Line 164: Why are these refractive index values different from those in Figure 137? Is it because of the wavelength? The wavelengths should be specified clearly in every instance.

**Response:** We thank the reviewer for pointing this out. The refractive index values differ from those in Line 195 because they correspond to different wavelengths. Here, our focus is on calculating the $E_{abs}$ at 630 nm, which corresponds to the measurement wavelength of the CAPS instrument. We apologize for any confusion this may have caused and have clarified the wavelengths explicitly in the main text (Lines 236-240):

"*Given the measurement data available in this study, the Core-shell Mie theory was used to calculate the $E_{abs}$ of BC-containing particles at a wavelength of 630 nm. The refractive index (RI) of BC and its coatings are assumed to be n=1.85+0.71i and n=1.5+0i at a wavelength of 630 nm (Liu et al., 2015; Liu et al., 2014).*"

**[Comment 19]** Figure 1: The core shell Mie model is for 200 nm, what is that a radius or a diameter, and is it mass equivalent, or something else? Also, why 200 nm exactly? More discussion on the model would be useful.

**Response:** We thank the reviewer for the comment. The 200 nm value in the Core-shell Mie model refers to the mass-equivalent diameter of the BC-containing particles, not the radius. The modeled $E_{abs}$ of 200 nm mass-equivalent diameter was selected to illustrate the pronounced increase of BC

$E_{abs}$ with increasing $M_R$ in the Core-shell Mie model. To further support this conclusion and demonstrate the robustness of the trend, we have additionally included simulation results for 100 nm mass-equivalent diameter particles in Figure. 1. And some explanatory text has also been added to the figure caption.

[Figure]

*Figure. 1 (a) The time series of BC concentrations, bulk-averaged $M_R$, measured $E_{abs}$, and the chemical components (including organics, nitrate, sulfate, ammonium and chloride) measured by TOF-ACSM X, as well as relatively humidity (RH). Shaded regions indicate different cases: light yellow for Case 1, blue-green for Case 2, and gray for Case 3. (b) Comparison of measured $E_{abs}$ in different observation periods and with previous studies (Peng et al., 2016; Cappa et al., 2012; Zhai et al., 2022; Cappa et al., 2019; Huang et al., 2024). The black solid line represents the fitted smoothing curve of bulk-averaged $M_R$ and measured $E_{abs}$, with the shaded area indicating the 95% confidence interval of the fit. (c) Mean 48-h back-trajectory simulations initialized at 100 m above ground level. The back trajectories were calculated using the Hybrid Single-Particle Lagrangian Integrated Trajectory (HYSPLIT) model driven by GDAS meteorological fields.*

**[Comment 20]** Line 215: I would not use the word "closely" here; the agreement is reasonable, but not that close.

**Response:** We thank the reviewer for the suggestion. This sentence has been revised in the

manuscript to replace "closely matches" with "reasonably agrees" (Lines 293-295).

*"On the other hand, in Case 2, the $E_{abs}$ calculated using the traditional core-shell Mie model ($E_{abs\_uniform}$) reasonably agrees with the $E_{abs\_measured}$, whereas Case 3 shows a slightly lower level of consistency."*

**[Comment 21]** Line 227: What does "at different period" refer to exactly? Perhaps "periods".

**Response:** We appreciate the reviewer's suggestion. The phrase "at different period" has been corrected to "at different periods" for grammatical accuracy, and we have clarified that it refers to different observation (or clean/polluted) periods in the revised manuscript (Lines 318-320).

*"Fig. 2a and 2b illustrated significant differences in the normalized number distribution of BC-containing particles at $M_P$ = 4.35 fg and 9.38 fg during different observation periods."*

**[Comment 22]** Line 232: Why log10?

**Response:** We appreciate the reviewer's insightful question. The logarithmic transformation of $M_R$ ($\log_{10}(M_R)$) was applied to reduce the strong skewness in the $M_R$ distribution, which typically spans several orders of magnitude. Using $\log_{10}(M_R)$ allows the data to approximate a normal distribution, making the standard deviation a more representative measure of heterogeneity. This approach is also consistent with previous studies (Fierce et al., 2020; Huang et al., 2024).

**[Comment 23]** Line 241-242: I am not sure what "due to stems" means.

**Response:** We thank the reviewer for pointing out this grammatical mistake. The phrase "due to stems from" was indeed redundant and has been corrected in the revised manuscript (Lines 336-337):

*"Such discrepancies likely due to the uniform core-shell model's simplified treatment of $M_R$ heterogeneity in BC (Romshoo et al., 2024; Wang et al., 2021b)."*

**[Comment 24]** Lines 264-265: I am not clear how these ranges are defined.

**Response:** We thank the reviewer for this suggestion. In this study, the $M_R$ range defining the "transition state" was determined based on the variation of $C_{sca\_measured}/C_{sca\_modeled}$ with $M_R$. Specifically, the lower boundary corresponds to the $M_R$ value at which the ratio begins to increase after an initial decrease, while the upper boundary corresponds to the $M_R$ value where the ratio approaches 1. We hope this explanation helps clarify our definition.

**[Comment 25]** Line 265: I do not recall the authors discussing the pollution regimes of the three cases. I would give that more emphasis early on because that is probably at the root of the observed behaviors.

**Response:** We thank the reviewer for this helpful comment. We would like to clarify that the pollution regimes of the three cases have been analyzed in both the beginning of Section 3 and in the concluding part of the discussion. Specifically, we examined the differences in air-mass origins and chemical composition for the three cases, as shown in Lines 265-282 and Lines 363-370:

Lines 265-282: *"To investigate the temporal evolution of BC coating, the observation period was classified into three cases based on the variation of $E_{abs\_measured}$, using $E_{abs}$=1.5 as the reference*

*threshold (Fig.1). Case 1 (September 3-23 and October 1-7, 2023) corresponds to $E_{abs\_measured}$ significantly below 1.5. These periods were characterized by relatively low non-refractory $PM_1$ concentrations and high wind speeds (WS = 0.94 ± 0.04 m/s). Back-trajectory analysis further shows that Case 1 was dominated by clean marine and nearby local air masses, resulting in relatively clean and weakly aged conditions (Fig. 1a). Case 2 (September 24–30) corresponds to periods when $E_{abs\_measured}$ remained continuously higher than 1.5. This episode occurred under stagnant meteorological conditions-characterized by weak winds (WS = 0.81 ± 0.02 m/s) and elevated relative humidity (RH = 81.34 ± 17.12%) -that favored secondary aerosol formation. Back-trajectory analysis further indicates that Case 2 was dominated by air masses transported from Jiangsu and passing through northern Zhejiang, enhancing pollutant accumulation and promoting more aged BC conditions. Case 3 includes periods when $E_{abs\_measured}$ persistently fluctuated around 1.5. The air masses during this period were a mixture of polluted inland outflow and clean marine inflow, suggesting the air masses were moderately aged–which explains the intermediate $E_{abs}$."*

Lines 363-370: "*The higher $M_R$ thresholds observed in Case 2 and Case 3 indicate that under polluted conditions, BC particles can reach an optically core-shell-like state with comparatively less coating material. This likely reflected accelerated aging driven by enhanced secondary formation and condensation of inorganics and organics on BC, facilitated by stagnant meteorological conditions (low wind speed). Such conditions promote efficient coating growth on BC-containing particles, strengthening their light-absorption capability and leading to high $E_{abs}$. Therefore, compared with Case 1, BC in Case 2 and Case 3 required less coating material to reach the core-shell configuration.*"

**[Comment 26]** Line 269: What do the authors mean by heterogeneous aging?

**Response:** We thank the reviewer for pointing out this unclear expression. We apologize for the ambiguity. In this context, "heterogeneous aging" refers to the evolution of heterogeneity among BC-containing particles during their aging process, such as differences in coating degrees or mixing states that develop over time. To avoid confusion, this part has been fully revised and integrated into other related sections. The updated content can be found in Lines 363-370 of the revised manuscript.

"*The higher $M_R$ thresholds observed in Case 2 and Case 3 indicate that under polluted conditions, BC particles can reach an optically core-shell-like state with comparatively less coating material. This likely reflected accelerated aging driven by enhanced secondary formation and condensation of inorganics and organics on BC, facilitated by stagnant meteorological conditions (low wind speed). Such conditions promote efficient coating growth on BC-containing particles, strengthening their light-absorption capability and leading to high $E_{abs}$. Therefore, compared with Case 1, BC in Case 2 and Case 3 required less coating material to reach the core-shell configuration.*"

**[Comment 27]** Line 272: While plausible, this explanation (on why "BC required less coating material to evolve into a core-shell structure") is not completely clear to me, and the authors might want to elaborate more.

**Response:** We thank the reviewer for this valuable comment and apologize for the lack of clarity in our previous explanation. We have revised this part to provide a more detailed and explicit discussion. The updated content can be found in Lines 367-370 of the revised manuscript.

"*The higher $M_R$ thresholds observed in Case 2 and Case 3 indicate that under polluted conditions,*

*BC particles can reach an optically core-shell-like state with comparatively less coating material. This likely reflected accelerated aging driven by enhanced secondary formation and condensation of inorganics and organics on BC, facilitated by stagnant meteorological conditions (low wind speed). Such conditions promote efficient coating growth on BC-containing particles, strengthening their light-absorption capability and leading to high $E_{abs}$. Therefore, compared with Case 1, BC in Case 2 and Case 3 required less coating material to reach the core-shell configuration."*

**[Comment 28]** Line 292: Again, I think "closely" is too strong.

**Response:** We have revised the sentence to avoid overstatement, changing "closely aligns with" to "agrees with." (Lines 389)

**[Comment 29]** Lines 300-302: I believe in the paper by Fierce et al., the discrepancy was suggested to be due to MR heterogeneity but mostly for Large coatings, while for low coatings the deviations were associated with non-core-shell configurations, so I do not see a clear discrepancy between the two studies, on the contrary, it seems to me that the results are quite similar when considered for similar coating regimes.

**Response:** We thank the reviewer for this comment. We agree that the interpretation in Fierce et al. (2020) depends on the coating conditions, and that our results are generally consistent with their findings when similar coating levels are considered. In our study, the relatively larger deviation observed for Case 1, compared to previous particle-resolved modeling studies, is primarily due to the smaller standard deviation of particle-to-particle $M_R$ in our observations, which leads to a greater discrepancy between measured and modeled $E_{abs}$. This explanation has been clarified in the revised manuscript (Lines 396-401).

*"The results show that for Case 1, although the discrepancy between the measured and modeled values exhibits a decreasing trend, the average deviation remains as high as 38%. This larger deviation, compared to previous particle-resolved modeling studies, is primarily attributed to the smaller dispersion of particle-to-particle $M_R$ observed in Case 1 (Fig. 2c) relative to their model simulations (Fierce et al., 2020)."*

**[Comment 30]** Figure 4(b), 4(c), and 4(d), what is on the x axis? Please put an axis title and units if applicable.

**Response:** We thank the reviewer for pointing this out. Axis labels and units have been added to Figures 4(b), 4(c), and 4(d) for clarity. Specifically, the x-axis represents $M_R$ ranges corresponding to "transition-state" BC-containing particles, which is now clearly indicated in the revised figures.

**[Comment 31]** Line 323: Why inorganic specifically? Why not also organics?

**Response:** We thank the reviewer for this comment. In response to feedback from multiple reviewers, we have thoroughly revised the discussion in this section to clarify the factors influencing $E_{abs}$ considering both inorganic and organic coatings. Our analysis indicates that among the different aerosol chemical components examined, the $E_{abs}$ increases systematically only with the increasing contribution of secondary nitrate, whereas variations in other components show no significant effect. These findings are now fully reflected in the revised manuscript, which provides a more comprehensive and data-supported interpretation (Lines 415-457).

"*The transitional-state particles are BC-containing particles in the process of evolving from loosely aggregated fractal-like structures toward quasi–core–shell configurations (Moffet et al., 2016; Moteki and Kondo, 2007). The abundance of transitional-state particles varies notably under different atmospheric conditions, directly influencing the measured $E_{abs}$ (Liu et al., 2017a). During clean days (Case 1), the atmospheric environment was characterized by low $PM_1$ concentrations, weak secondary formation, and highly variable coating conditions. Under such conditions, our measurements show that BC-containing particles were dominated by transitional-state structures (Fig. S9), representing the intermediate stage between externally mixed aggregates and fully developed quasi–core–shell structures. The limited and heterogeneous coating distribution on these particles substantially weakens the lensing effect, resulting in lower measured $E_{abs}$ (Peng et al., 2016). Because the core-shell Mie model inherently assumes a uniform and concentric coating, it does not accurately represent the optical behavior of these transitional particles, leading to a pronounced overestimation of measured $E_{abs}$ during Case 1. This indicates that, under clean conditions, the optical properties of transitional-state particles are the key driver of the model-observation discrepancy. In contrast, the haze period (Case 2) represents a more aged and heavily coated aerosol environment and provides a useful reference for understanding the factors influencing the measured $E_{abs}$. During Case 2, the high aerosol loading and elevated bulk-averaged $M_R$ were largely influenced by regional transport, as air masses at 100 m, 500 m, and 1000 m all followed similar pathways from the northern Yangtze River Delta into northern Zhejiang (Fig. 1c and Fig. S7). Stagnant meteorological conditions, elevated relative humidity, and enhanced oxidative capacity further facilitated vigorous liquid-phase and photochemical reactions, promoting the abundant formation of secondary coatings on BC surfaces (Peng et al., 2016). Notably, our observations show that $E_{abs}$ increases systematically with the increasing contribution of secondary nitrate (Fig. S8), consistent with the fact that nitrate-rich conditions enhance aqueous-phase oxidation and accelerate the formation of thick inorganic coatings (Liu et al., 2017b). As a result, a much larger fraction of BC-containing particles exhibited internally mixed, quasi–core-shell structures rather than transitional states (Fig. S9), which explains why the core-shell Mie model performs substantially better for Case 2 than for Case 1. This contrast reinforces the central role of transitional-state particles in determining measured $E_{abs}$ when coatings are sparse, irregular, or partially developed. Given the strong influence of transitional-state particles on measured $E_{abs}$ in Case 1, precise constraints on their optical behavior are crucial for improving $E_{abs}$ estimates across different atmospheric scenarios. To address this, an empirical formula based on optical measurements was developed to estimate the $E_{abs}$ of BC-containing particles in the "transition state", derived from fitting the measured $C_{sca}$ against $M_R$ (Fig. 3d-3f). By applying this empirical formula to the calculation of $E_{abs}$, the resulting value for Case 1 was 1.21 ± 0.01. For Case 2 and Case 3, the $E_{abs}$ calculated using the same formula ($E_{abs\_param}$) remained slightly lower than the $E_{abs\_measured}$, but the deviation was within 20%, demonstrating the reliability of the approach across different atmospheric conditions.*"

**[Comment 32-33]** Line 337: The authors should clearly describe the "improved method". Here, it is not clear at all what that is. How was the fit performed, and what did they do with the fit?

Lines 360 to 368: How do the authors suggest developing such a unified model?

**Response:** We thank the reviewer for these comments. Both points regarding the "improved method" and the development of a unified model have been fully addressed in **[Comment 5]**, where we

provide a detailed description of the observationally constrained parameterization, including how the $C_{sca\_measured}$-$M_R$ fit is performed, how the fit is used to estimate $E_{abs}$ for transition-state particles, and how the framework can be adapted for different atmospheric conditions. Please refer to **[Comment 5]** for the complete explanation.

**[Comment 34]** Figure S2: It is not clear how the CPC would measure a size distribution, being a simple counter. The caption mentions a DMA; perhaps that should be clearer from the legend. What are SCHG and BBHG (scattering and broadband high gain?). Spell out acronyms in the caption.

**Response:** We thank the reviewer for this helpful comment. We have clarified in the figure caption that the size distribution was measured using a DMA coupled with a CPC. In addition, the acronyms SCHG and BBHG have been spelled out as scattering high gain and broadband high gain, respectively. The legend and caption have been revised in revised Supporting Information.

[Figure]

***Figure S2****. The number concentration measured by SP2 and CPC after DMA classification of size-resolved Aquadag aerosols (a). (b) showed the calibration factor for scattering high gain (SCHG) before and after campaign. (c) and (d) display the correlation between incandescence peak height and BC particle mass at broadband incandescence high gain (BBHG) and broadband incandescence low gain (BBLG), respectively.*

**[Comment 35]** Figures S4 and S5: the signal ranges are very different. Why is that? Can the nephelometer and the CAPS measure scattering coefficients below 0.01 Mm-1 accurately and precisely? That is hard to believe. Also how were the Mie simulations being informed, in terms of concentrations, size, and index of refraction?

**Response:** We thank the reviewer for these valuable comments. The apparent difference in signal ranges between Figures S4 and S5 was due to a labeling error in Figure S5. The unit should be scattering cross section ($m^2$) rather than scattering coefficient ($Mm^{-1}$), and this has been corrected

in the revised Supplementary Information.

**[Comment 36]** Figure S7: What does the horizontal pink dashed line indicate, the mean Eabs?

**Response:** We thank the reviewer for the comment. The horizontal pink dashed lines represent the upper and lower bounds of the 90% confidence interval of the fitted curve, which were automatically calculated by Igor based on the fit. This has been clarified in the figure caption in the revised Supplementary Information (see Figure S6).

**[Comment 37]** Figure S8: Some more information is needed. How were the trajectories being calculated? What is the time span, what is the elevation used, can the authors also provide vertical trajectories to see whether the air masses are coming from the boundary layer or from aloft to understand if it is near or long-range transportation?

**Response:** We thank the reviewer for this comment. To improve clarity, we have merged the original Figure S8 with Figure 1 in the revised manuscript (see **[Comment 19]**). The back trajectories were calculated using the HYSPLIT model driven by GDAS meteorological fields. In the updated Figure 1, we show 48-hour backward trajectories initialized at 100 m above ground level. To further examine the vertical representativeness and to assess whether the air masses originate from the boundary layer or from higher altitudes, additional trajectories initialized at 500 m and 1000 m (also for 48 hours) are now provided in Figure S7. These additional simulations demonstrate that the air-mass pathways are similar across different starting altitudes, supporting our interpretation of the transport characteristics.

[Figure]

***Figure S7.*** *48-hour backward trajectories of air masses during all observation periods at 100 m, 500 m, and 1000 m above ground level, calculated using the Hybrid Single-Particle Lagrangian Integrated Trajectory (HYSPLIT) model driven by GDAS meteorological fields.*

**[Comment 38]** Figure S9: Are these based on the SP2?

**Response:** We thank the reviewer for the comment. Yes, the data in Figure S9 are based on SP2 measurements, and this has been explicitly indicated in the figure title in the revised Supplementary Information (Figure S9).

**[Comment 39]** Figure S10: The units on the x-axis are not critical, but still, it would be nice to have

them.

**Response:** We thank the reviewer for the comment. In response to this and other reviewers' suggestions, we have supplemented the distributions of scattering signals for all $M_p$ and added the units on the x-axis. The updated figure is now presented as Figure S11 in the revised Supplementary Information.

[Figure]

***Figure S11****. Multiple charging diagnostics in the tandem CPMA-SP2 system. The number distribution of scattering-signal peak heights at each selected $M_p$ was fitted using a bimodal Gaussian function, with one peak representing singly charged particles and the other corresponding to doubly charged particles. The intersection point of the two fitted peaks was used as the threshold for distinguishing singly from doubly charged particles at that $M_p$, enabling a quantitative evaluation of multiple charging effects.*

**[Comment 40]** Figure S11: Units on x axes (fg?).

**Response:** In response to the comment, the units on the x-axis have been added as fg. The updated figure is now presented as Figure S12 in the revised Supplementary Information.

**[Comment 41]** Some limited references that might be of relevance:
1. Beeler, P., et al., Light absorption enhancement of black carbon in a pyrocumulonimbus cloud. Nature Communications, 2024. 15(1): p. 6243.
2. Ueda, S., et al., Light absorption and morphological properties of soot-containing aerosols observed at an East Asian outflow site, Noto Peninsula, Japan. Atmos. Chem. Phys., 2016. 16(4): p. 2525-2541.
3. China, S., et al., Morphology and mixing state of individual freshly emitted wildfire carbonaceous particles. Nature Communications, 2013. 4.

4. Cross, E.S., et al., Soot Particle Studies—Instrument Inter-Comparison—Project Overview. Aerosol Science and Technology, 2010. 44(8): p. 592-611.

5. Corbin, J.C., R.L. Modini, and M. Gysel-Beer, Mechanisms of soot-aggregate restructuring and compaction. Aerosol Science and Technology, 2022: p. 1-48.

6. Adachi, K. and P.R. Buseck, Changes of ns-soot mixing states and shapes in an urban area during CalNex. Journal of Geophysical Research: Atmospheres, 2013. 118(9): p. 3723–3730.

7. Adachi, K., S.H. Chung, and P.R. Buseck, Shapes of soot aerosol particles and implications for their effects on climate. Journal of Geophysical Research-Atmospheres, 2010. 115.

8. Adachi, K., et al., Mixing states of light-absorbing particles measured using a transmission electron microscope and a single-particle soot photometer in Tokyo, Japan. Journal of Geophysical Research: Atmospheres, 2016. 121(15): p. 9153-9164.

9. leviChang, H. and T.T. Charalampopoulos, Determination of the Wavelength Dependence of Refractive-Indexes of Flame Soot. Proceedings of the Royal Society-Mathematical and Physical Sciences, 1990. 430(1880): p. 577-591.

10. Moteki, N., Measuring the complex forward-scattering amplitude of single particles by self-reference interferometry: CAS-v1 protocol. Optics Express, 2021. 29(13): p. 20688-20714.

11. Liu, S., et al., Enhanced light absorption by mixed source black and brown carbon particles in UK winter. Nat Commun, 2015. 6.

**Response:** We thank the reviewer for providing these references. The manuscript has been thoroughly revised, and most of the suggested references have already been incorporated. A few references (specifically references 4, 9 and 10) were not cited, as their focus is not directly aligned with the objectives of the present study.

**References:**

Adachi, K. and Buseck, P. R.: Changes of ns-soot mixing states and shapes in an urban area during CalNex, Journal of Geophysical Research-Atmospheres, 118, 3723-3730, 10.1002/jgrd.50321, 2013.

Adachi, K., Chung, S. H., and Buseck, P. R.: Shapes of soot aerosol particles and implications for their effects on climate, Journal of Geophysical Research: Atmospheres, 115, D15206, 10.1029/2009jd012868, 2010.

Brooks, J., Liu, D., Allan, J. D., Williams, P. I., Haywood, J., Highwood, E. J., Kompalli, S. K., Babu, S. S., Satheesh, S. K., Turner, A. G., and Coe, H.: Black carbon physical and optical properties across northern India during pre-monsoon and monsoon seasons, Atmospheric Chemistry and Physics, 19, 13079-13096, 10.5194/acp-19-13079-2019., 2019.

Cappa, C. D., Zhang, X., Russell, L. M., Collier, S., Lee, A. K. Y., Chen, C. L., Betha, R., Chen, S., Liu, J., Price, D. J., Sanchez, K. J., McMeeking, G. R., Williams, L. R., Onasch, T. B., Worsnop, D. R., Abbatt, J., and Zhang, Q.: Light Absorption by Ambient Black and Brown Carbon and its Dependence on Black Carbon Coating State for Two California, USA, Cities in Winter and Summer, Journal of Geophysical Research: Atmospheres, 124, 1550-1577, 10.1029/2018JD029501, 2019.

Cappa, C. D., Onasch, T. B., Massoli, P., Worsnop, D. R., Bates, T. S., Cross, E. S., Davidovits, P., Hakala, J., Hayden, K. L., Jobson, B. T., Kolesar, K. R., Lack, D. A., Lerner, B. M., Li, S.-M., Mellon, D., Nuaaman, I., Olfert, J. S., Petäjä, T., Quinn, P. K., Song, C., Subramanian, R., Williams, E. J., and Zaveri, R. A.: Radiative Absorption Enhancements Due to the Mixing State of Atmospheric Black

Carbon, Science, 337, 1078-1081, https://doi.org/10.1126/science.1223447, 2012.

China, S., Mazzoleni, C., Gorkowski, K., Aiken, A. C., Dubey, M. K., and Michigan Technological Univ, H. M. I.: Morphology and mixing state of individual freshly emitted wildfire carbonaceous particles, Nature Communications, 4, 2122-2122, 10.1038/ncomms3122, 2013.

Corbin, J. C., Modini, R. L., and Gysel-Beer, M.: Mechanisms of soot-aggregate restructuring and compaction, Aerosol Science and Technology, 57, 89-111, 10.1080/02786826.2022.2137385, 2023.

Fierce, L., Onasch, T. B., Cappa, C. D., Mazzoleni, C., China, S., Bhandari, J., Davidovits, P., Al Fischer, D., Helgestad, T., Lambe, A. T., Sedlacek, A. J., Smith, G. D., Wolff, L., Brookhaven National Lab, U. N. Y., and Pacific Northwest National Lab, R. W. A.: Radiative absorption enhancements by black carbon controlled by particle-to-particle heterogeneity in composition, Proceedings of the National Academy of Sciences, 117, 5196-5203, 10.1073/pnas.1919723117, 2020.

Gao, R. S., Schwarz, J. P., Kelly, K. K., Fahey, D. W., Watts, L. A., Thompson, T. L., Spackman, J. R., Slowik, J. G., Cross, E. S., Han, J. H., Davidovits, P., Onasch, T. B., and Worsnop, D. R.: A Novel Method for Estimating Light-Scattering Properties of Soot Aerosols Using a Modified Single-Particle Soot Photometer, Aerosol Science and Technology, 41, 125-135, 10.1080/02786820601118398, 2007.

Huang, X.-F., Peng, Y., Wei, J., Peng, J., Lin, X.-Y., Tang, M.-X., Cheng, Y., Men, Z., Fang, T., Zhang, J., He, L.-Y., Cao, L. M., Liu, C., Zhang, C., Mao, H., Seinfeld, J. H., and Wang, Y.: Microphysical complexity of black carbon particles restricts their warming potential, One Earth, 7, 10.1016/j.oneear.2023.12.004, 2024.

Liu, D., Taylor, J. W., Young, D. E., Flynn, M. J., Coe, H., and Allan, J. D.: The effect of complex black carbon microphysics on the determination of the optical properties of brown carbon: BC morphology on BrC optical properties, Geophysical Research Letters, 42, 613-619, 10.1002/2014GL062443, 2015.

Liu, D., Allan, J. D., Young, D. E., Coe, H., Beddows, D., Fleming, Z. L., Flynn, M. J., Gallagher, M. W., Harrison, R. M., Lee, J., Prevot, A. S. H., Taylor, J. W., Yin, J., Williams, P. I., and Zotter, P.: Size distribution, mixing state and source apportionment of black carbon aerosol in London during wintertime, Atmospheric Chemistry and Physics, 14, 10061-10084, 10.5194/acp-14-10061-2014, 2014.

Liu, D., Whitehead, J., Alfarra, M. R., Reyes-Villegas, E., Spracklen, D. V., Reddington, C. L., Kong, S., Williams, P. I., Ting, Y.-C., Haslett, S., Taylor, J. W., Flynn, M. J., Morgan, W. T., McFiggans, G., Coe, H., and Allan, J. D.: Black-carbon absorption enhancement in the atmosphere determined by particle mixing state, Nature Geoscience, 10, 184-188, 10.1038/ngeo2901, 2017a.

Liu, H., Pan, X., Liu, D., Liu, X., Chen, X., Tian, Y., Sun, Y., Fu, P., and Wang, Z.: Mixing characteristics of refractory black carbon aerosols at an urban site in Beijing, Atmospheric Chemistry and Physics, 20, 5771-5785, 10.5194/acp-20-5771-2020, 2020.

Liu, Y., Wu, Z., Wang, Y., Xiao, Y., Gu, F., Zheng, J., Tan, T., Shang, D., Wu, Y., Zeng, L., Hu, M., Bateman, A. P., and Martin, S. T.: Submicrometer Particles Are in the Liquid State during Heavy Haze Episodes in the Urban Atmosphere of Beijing, China, Environmental Science & Technology Letters, 4, 427-432, 10.1021/acs.estlett.7b00352, 2017b.

Moffet, R. C., O'Brien, R. E., Alpert, P. A., Kelly, S. T., Pham, D. Q., Gilles, M. K., Knopf, D. A., Laskin, A., Pacific Northwest National Lab, R. W. A. E. M. S. L., Lawrence Berkeley National Lab, B. C. A., and Stony Brook Univ, S. B. N. Y.: Morphology and mixing of black carbon particles collected in central California during the CARES field study, Atmospheric chemistry and physics, 16, 14515-

14525, 10.5194/acp-16-14515-2016, 2016.

Moteki, N. and Kondo, Y.: Effects of Mixing State on Black Carbon Measurements by Laser-Induced Incandescence, Aerosol Science and Technology, 41, 398-417, 10.1080/02786820701199728, 2007.

Peng, J., Hu, M., Guo, S., Du, Z., Zheng, J., Shang, D., Zamora, M. L., Zeng, L., Shao, M., Wu, Y.-S., Zheng, J., Wang, Y., Glen, C. R., Collins, D. R., Molina, M. J., and Zhang, R.: Markedly enhanced absorption and direct radiative forcing of black carbon under polluted urban environments, Proceedings of the National Academy of Sciences, 113, 4266-4271, 10.1073/pnas.1602310113, 2016.

Romshoo, B., Müller, T., Ahlawat, A., Wiedensohler, A., Haneef, M. V., Imran, M., Warsi, A. B., Mandariya, A. K., Habib, G., and Pöhlker, M. L.: Significant contribution of fractal morphology to aerosol light absorption in polluted environments dominated by black carbon (BC), npj Climate and Atmospheric Science, 7, 87, 10.1038/s41612-024-00634-0, 2024.

Wang, T. T., Zhao, G., Tan, T. Y., Yu, Y., Tang, R. Z., Dong, H. B., Chen, S. Y., Li, X., Lu, K. D., Zeng, L. M., Gao, Y. Q., Wang, H. L., Lou, S. R., Liu, D. T., Hu, M., Zhao, C. S., and Guo, S.: Effects of biomass burning and photochemical oxidation on the black carbon mixing state and light absorption in summer season, Atmospheric Environment, 248, 10.1016/j.atmosenv.2021.118230, 2021a.

Wang, Y., Li, W., Huang, J., Liu, L., Pang, Y., He, C., Liu, F., Liu, D., Bi, L., Zhang, X., and Shi, Z.: Nonlinear Enhancement of Radiative Absorption by Black Carbon in Response to Particle Mixing Structure, Geophysical Research Letters, 48, 10.1029/2021gl096437, 2021b.

Zhai, J., Yang, X., Li, L., Bai, B., Liu, P., Huang, Y., Fu, T.-M., Zhu, L., Zeng, Z., Tao, S., Lu, X., Ye, X., Wang, X., Wang, L., and Chen, J.: Absorption Enhancement of Black Carbon Aerosols Constrained by Mixing-State Heterogeneity, Environmental Science & Technology, 56, 1586-1593, 10.1021/acs.est.1c06180, 2022.

Zhang, F., Shen, J., Xu, D., Shen, J., Qin, Y., Shi, R., Wei, J., Xu, Z., Pei, X., Tang, Q., Chen, H., Xu, B., and Wang, Z.: Unveiling the key drivers and formation pathways for secondary organic aerosols in an eastern China megacity, Journal of Hazardous Materials, 498, 10.1016/j.jhazmat.2025.139925, 2025.

Zhang, Y., Zhang, Q., Cheng, Y., Su, H., Kecorius, S., Wang, Z., Wu, Z., Hu, M., Zhu, T., Wiedensohler, A., and He, K.: Measuring the morphology and density of internally mixed black carbon with SP2 and VTDMA: new insight into the absorption enhancement of black carbon in the atmosphere, Atmospheric Measurement Techniques, 9, 1833-1843, 10.5194/amt-9-1833-2016, 2016.

Zhang, Y. X., Zhang, Q., Yao, Z. L., and Li, H. Y.: Particle Size and Mixing State of Freshly Emitted Black Carbon from Different Combustion Sources in China, Environmental Science & Technology, 54, 7766-7774, 10.1021/acs.est.9b07373, 2020.

Zhao, G., Shen, C., and Zhao, C.: Technical note: Mismeasurement of the core-shell structure of black carbon-containing ambient aerosols by SP2 measurements, Atmospheric Environment, 243, 117885, 10.1016/j.atmosenv.2020.117885, 2020.

---

## Author Comment (AC2)

**Response to Reviewer #2:**

The manuscript by Jing Wei et al. presents [charger-]CPMA-SP2 measurements, from which measurements of the distribution of rBC fraction can be inferred. The manuscript is carefully written and shows that the authors have thought carefully about the interpretation of their results. The literature context is good, and the data set represents a significant contribution to the literature. However, there is one major gap which must be addressed before this manuscript is published (multiple charge correction) and another big opportunity for improvement (compare the SP2 with the analytical grade nephelometers, not only with the SP2-LEO analysis). There are also a few smaller opportunities for improvement, described below.

**[Comment 1]** The major gap is that this work does not cite, nor implement, the advances in [charger-]CPMA-SP2 data inversion which have been published since the original work by Liu et al., 2017 (cited by the authors). Those advances have been published in a series of papers, most recently Naseri et al. 2024, which are accompanied by open-source code for performing the calculations. The key feature of these calculations is to acknowledge the aerosol bipolar charger, which is necessarily placed before the CPMA to make sense of its output. Without this charger, the SP2 would see a large fraction of neutral particles downstream of the CPMA. With this charger, the SP2 would see not only the desired particles (q=1, the CPMA setpoint) but also a large fraction of doubly (q=2) and triply (q=3) charged particles. The purpose of the CPMA-SP2 data inversion code cited above is the account for these q>1 particles. For example, if the CPMA was set to 2 fg/e (2 femtograms per charge) and the SP2 observed a particle with 1 fg of rBC, that particle might be a 50%/50% rBC/coating particle (q=1, 2fg/1e), a 25%/75% rBC/coating particle (q=2, 4fg/2e), or even a 16% rBC (6fg/3e). A full data inversion will account for this.
Implementing this data inversion would affect:
- Eq 1, the authors write "without any assumptions", but there is an assumption of singly charged particles.
- All figures, which currently do not distinguish "doubly charged particle with BC fraction 0.5" from "singly charged particle with half the coating".

**Response:** We thank the reviewer for this comment. We would like to clarify that the influence of multiply charged particles (q > 1) has been explicitly considered in our CPMA-SP2 measurements. An X-ray aerosol neutralizer (TSI 3088) was installed upstream of the CPMA to charge the aerosols before entering the classifier (Lines 120-123). Although we did not directly use the inversion code from Liu et al. (2017a) or Naseri et al. (2024), we have applied an independent multiple charge correction procedure to the data, allowing us to identify and select only singly charged particles. All relevant details had already been included in Text S1 and Figure S10 and Figure S11. Besides, we have now explicitly stated the assumption of singly charged particles before Eq. 1 in the revised manuscript (Lines 176-178).

To facilitate readers' understanding of the data correction procedures and the CPMA–SP2 system configuration, we have added the relevant descriptions in the Methods (Lines 135-138) and Text S1 and included the aerosol neutralizer in the schematic diagram of Figure S1.

Lines 120-123: "*In this setup, particles with known mass ($M_p$) selected by CPMA were injected into the SP2. An X-ray aerosol neutralizer (TSI 3088) was installed upstream of the CPMA to charge the*

*aerosols to a Boltzmann equilibrium before entering the classifier.*"

Lines 176-178: "*Under the assumption of singly charged particles, the mixing state of a single BC-containing particle can be represented by the mass ratio of the BC coating to the BC core, without relying on assumptions about particle morphology or coating structure,*"

Lines 135-138: "*In the subsequent data processing, measurements from the CPMA-SP2 system were first corrected for multiple charging effects. Additional corrections, including transfer function, detection efficiency, and time delay, were also applied, as detailed in the Text S1.*"

Other major comments

**[Comment 2]** The MAC vs M_R plot of Figure S6 shows that MAC did not increase even at M_R of 5.5. Mie theory and lab experiments show that coated soot has E_abs 2.0 at M_R of 5.5 (Cappa et al., 2012; doi:10.1126/science.1223447). So the fundamental premise of this paper seems to be violated by Figure S6.

**Response:** We thank the reviewer for this comment. We would like to clarify that the MAC vs. $M_R$ relationship in Figure S6 reflects ambient measurements under real atmospheric conditions, which are more complex than idealized laboratory experiments or theoretical Mie calculations. In the atmosphere, factors such as particle mixing state, coating composition, hygroscopic growth, and measurement uncertainties can influence the observed MAC values. Thus, the lack of further MAC enhancement beyond $M_R$=5 in our field measurement does not violate laboratory findings.

**[Comment 3]** The abstract claims a morphology dependent correction scheme but none of the data are morphology dependent. The introduction could be enhanced by adding a discussion of how morphology can be measured. Or, the word morphology could be removed from the final paragraph. I suggest removing it, as the true measurement here is M_R, not morphology.

**Response:** We thank the reviewer for this constructive comment and agree that references to particle morphology were not appropriate. All morphology-related descriptions have been removed from the Abstract and main text. These statements have been revised to explicitly describe $M_R$-dependent optical transition behavior, rather than particle morphology. We believe these changes improve the accuracy and clarity of the revised manuscript.

**[Comment 4]** Line 154, the MAC of 9 m2/g is 3 standard deviations than the expected value of 8 +- 0.7 m2/g at 550 nm (Liu et al 2020). Please comment.

**Response:** We thank the reviewer for this comment. The reported MAC of 9.08 $m^2$/g corresponds to pure BC, representing the intrinsic light absorption efficiency of BC without the influence of any coating materials. Therefore, this value is independent of the mixing state and reflects the inherent optical property of BC itself. The slightly higher value compared to the expected $8 \pm 0.7$ $m^2$/g at 550 nm (Liu et al., 2020) can be attributed to several plausible factors. (1) BC morphology: even for uncoated BC, variations in aggregate compactness, fractal dimension, and cluster structure can modulate light absorption efficiency. (2) Instrumental and calibration effects: Differences in SP2 calibration with Aquadag standards, detector response, and limited sampling statistics can cause small biases in the retrieved MAC. Overall, the measured MAC of pure BC in this study is considered more representative of the local BC characteristics under actual atmospheric conditions.

In response to similar comments from other reviewers, we have also added a brief note in the main manuscript to clarify these uncertainties (Lines 219-228).

"*The MAC$_{BC\_core\_measured}$ at wavelength of 630 nm was 9.08 ± 0.53 m$^2$ g$^{-1}$ (mean ± 90% confidence Interval) (Fig. S6). Based on our error propagation analysis, which accounts for measurement uncertainties in particle absorption and BC mass as well as the standard error of the extrapolation, the estimated uncertainty of MAC$_{BC\_core\_measured}$ is approximately 19-23% (Text S2). And the uncertainty of E$_{abs\_measured}$ is approximately 26 - 32% (Text S2). For comparison, the MAC$_{BC\_core\_measured}$ is slightly higher than the value of 7.5 m$^2$ g$^{-1}$ recommended by Bond and Bergstrom (2006) but still within the range reported by other study (~6.5 - 17 m$^2$ g$^{-1}$) (Zanatta et al., 2016), likely due to variations in measurement methods, and site-specific atmospheric conditions.*"

**[Comment 5]** Figure 3: Is the positive correlation truly statistically significant? Please use prediction bands and add uncertainty in the fit coefficient. Is it truly different from 1? It seems to me that there is not a significant relationship.

**Response:** We appreciate the reviewer's comment. We have re-examined the correlation between the measured and modeled scattering cross-sections and performed a statistical significance test for the fitted slope. The P-value of the regression coefficient is less than 0.05, indicating that the positive correlation is statistically significant. In addition, we have added the 95% prediction bands and the uncertainty of the fitted slope in Figure 3 to better illustrate the confidence range of the relationship. We hope that these additional analyses and revisions provide a more complete and convincing presentation of the relationship.

[Figure]

***Figure. 3*** *The optical behavior of BC-containing particles as a function of mass ratio (coating-to-*

*BC, $M_R$). **(a)** – **(c)** show the ratio of measured to modeled scattering cross-section at the wavelength of 1064 nm during different cases. The shadows indicate the $M_R$ ranges corresponding to "transition-state" BC-containing particles. **(d)** – **(f)** present the measured scattering cross-section as a function of $M_R$, along with fitted optical transition-dependent models representing the "transition-state" BC-containing particles. The fitted curves include the corresponding P-values, and the shaded areas denote the 95% confidence intervals.*

**[Comment 6]** Line 181 why not model the nephelometer instead of the SP2? You have already assumed a wavelength independent rBC refractive index.

**Response:** We appreciate the reviewer's suggestion. However, the nephelometer measures the bulk scattering coefficient of all particles in the ambient, rather than the scattering cross section ($C_{sca}$) of BC-containing particles. In contrast, the SP2 provides single-particle information, allowing us to quantify the scattering cross section of individual BC-containing particles. Our goal in this study is to establish the relationship between the measured and modeled $C_{sca}$ of BC-containing particles as a function of the $M_R$, which reflects $M_R$-dependent optical transition behavior during BC aging. Therefore, only SP2-derived single-particle $C_{sca}$ are suitable for this analysis, while nephelometer measurements cannot serve this purpose. In this work, the nephelometer data were used only to compare with the CAPS measurements to evaluate the accuracy of the CAPS-derived scattering coefficients.

**[Comment 7]** Figure 4 shows "E_abs,improved" but this paper does not actually demonstrate an improvement. The results here do not seem generalizable. Rename to "E_abs,this work"

**Response:** We thank the reviewer for the suggestion. We agree that the term "E_abs, improved" may overstate the generality of our results. However, since the parameter represents the $E_{abs}$ derived from our refined modeling approach, we have renamed it as "$E_{abs\_param}$" throughout the text and Figure 4. This revised label more accurately reflects that it is based on an improved method rather than implying universally better performance.

Minor comments

**[Comment 8]** Line 56, you might cite Radney et al. 2014 and Corbin et al. 2023.

**Response:** We thank the reviewer for the helpful suggestion. The references Radney et al. (2014) and Corbin et al. (2023) have been added at Lines 70 to support the corresponding statement.

**[Comment 9]** Line 58, please cite a paper for "or located on the particle surface" or for the entire sentence.

**Response:** We thank the reviewer for the suggestion. The statement has been supported by appropriate references, and Zhang et al. (2008) and Adachi and Buseck (2013) have been added accordingly. (Lines 70-72)

*"Early aging stage feature uneven coatings, while aged particles show BC core either encapsulated or located near the particle surface (Zhang et al., 2008; Adachi and Buseck, 2013)."*

**[Comment 10]** Line 99, "noisy scattering signals" or "noisy incandescence signals"? The section is discussing M_R? Please also explain "noisy" with a few more words.

**Response:** We thank the reviewer for the comment. The sentence has been clarified to specify that the exclusion was based on noisy scattering signals, which were caused by weak signal intensity and low signal-to-noise ratio for small particles. The corresponding revision has been made in the revised manuscript. (Lines 130-133)

"*During further data analysis, particles with $M_P = 0.93$ fg and $M_P = 1.37$ fg exhibited excessively noisy scattering signals, likely due to weak signal intensity and low signal-to-noise ratio for small particles, and were therefore excluded from subsequent statistical analysis.*"

**[Comment 11]** Line 103, does this aquadag correction factor trace back to Gysel et al. (2011)? Either way, that early paper should be cited.

**Response:** We thank the reviewer for the helpful suggestion. The Aquadag correction factor used in this study is mainly based on the approach described by Liu et al. (2014) while Gysel et al. (2011) is now also cited to acknowledge the original development of this correction method. Both references have been added at Line 138-141 in the revised manuscript.

"*The mass of each BC core ($M_{BC\_core}$) was then calculated from the SP2 incandescence signal using the calibration described above, with a correction factor of 0.75 applied to the peak height (Liu et al., 2020; Liu et al., 2014; Zhang et al., 2018; Gysel et al., 2011).*"

**[Comment 12]** Line 124, why compare 2 difference averaging times?

**Response:** We thank the reviewer for the helpful comment. The detection limits of the Nephelometer and CAPS were reported according to their respective instrument specifications, which use different default averaging times. Our intention was not to directly compare the two instruments, but only to indicate their detection limits. This clarification has been added to the revised manuscript (Lines 171-173).

"*The lower detection limit of the Nephelometer at all three wavelengths was 0.3 $Mm^{-1}$ with a 60-second integration time, while that of the CAPS-ALB was 1 $Mm^{-1}$ with 30-second integration time.*"

**[Comment 13]** Line 125, Did you calibrate the CAPS extinction measurement with this scattering calibration?

**Response:** We thank the reviewer for this question. We thank the reviewer for this question. The CAPS extinction channel was not directly calibrated using the PSL scattering calibration. The CAPS was calibrated following the manufacturer-recommended procedure, in which extinction is referenced to Rayleigh scattering by filtered air. For monodisperse PSL particles, the optical extinction is theoretically equal to the scattering because PSL is a purely scattering aerosol. Therefore, the PSL-based scattering calibration was used as an independent verification of the CAPS extinction response rather than as its primary calibration. The good agreement between PSL-derived scattering (equivalent to extinction for PSL) and the CAPS extinction measurement confirms the accuracy and wavelength consistency of the CAPS calibration (Figure R1).

[Figure]

*Figure R1. PSL-derived scattering versus CAPS extinction measurements during PSL injection.*

**[Comment 14]** Line 137, this refractive index is very precise (1.48) would you not expect variability (1.5 to 1.6?). Also, a different RI is cited later on line 164.

**Response:** We thank the reviewer for this comment. In the revised manuscript, we have clarified the use of refractive indices in relation to the $M_R$-dependent optical transition behavior of BC-containing particles. At Line 137, the refractive indices were selected specifically for deriving the $M_R$-dependent optical transition behavior from SP2 scattering measurements at 1064 nm. The BC core is assigned a complex refractive index of 2.26-1.26i, while the non-absorbing coating has a refractive index of 1.48 and a density of 1.5 g cm$^{-3}$, values that have been widely used in previous studies (Liu et al., 2017a; Zhao et al., 2020; Liu et al., 2015). Besides, since the coating in this work does not absorb at 1064 nm, its absorption can be safely ignored for determining scattering and transition characteristics. In contrast, at line 164, our analysis focuses on calculating $E_{abs}$ at 630 nm, consistent with the CAPS measurement wavelength. We apologize for the confusion caused by the earlier description. We have now clarified and reorganized the wavelength-specific assumptions and the $M_R$-dependent optical transition behavior in the revised manuscript (Lines 189-200 and Lines 235-239):

Lines 190-200: *"The $M_R$-dependent optical transitions of BC-containing particles were further derived from SP2 measurements at a wavelength of 1064 nm. In the CPMA-SP2 system, when both $M_p$ and $M_{BC\_core}$ are known, the modeled scattering cross section ($C_{sca\_modeled}$) of BC-containing particles can be derived using Mie theory (Wang et al., 2021). This calculation assumes a core-shell structure, with the BC core having a refractive index of 2.26-1.26i (Liu et al., 2017a; Zhao et al., 2020) and the non-absorbing coating characterized by a refractive index of 1.48 and a density of 1.5 g cm$^{-3}$ at a wavelength of 1064 nm (Liu et al., 2015). The measured scattering cross section ($C_{sca\_measured}$) was obtained from the SP2 using the leading-edge-only (LEO) technique, which reconstructs the scattering signal as BC-containing particles pass through the SP2 laser beam due to partial evaporation of refractory-absorbing material."*

Lines 236-240: *"Given the measurement data available in this study, the Core-shell Mie theory was used to calculate the $E_{abs}$ of BC-containing particles at a wavelength of 630 nm. The refractive index (RI) of BC and its coatings are assumed to be n=1.85+0.71i and n=1.5+0i at a wavelength of 630 nm (Liu et al., 2015; Liu et al., 2014)."*

**[Comment 15]** Line 157, please add Generalized Mie Model to the list.

**Response:** We thank the reviewer for the suggestion. The Generalized Mie Model has been added to the list of commonly used models for calculating the optical properties of BC-containing particles (Line 232).

**[Comment 16]** Several examples: Please change "(Method)" to a section reference.

**Response:** We thank the reviewer for the suggestion. All instances of "(Method)" have been updated to refer to the specific section, Section 2.2, in the revised manuscript (Line 242).

**[Comment 17]** Line 181, mathematical -> empirical

**Response:** Corrected.

**[Comment 18]** Line 255-256, is there any experimental evidence to quantify the M_R the rBC would become fully embedded?

**Response:** We thank the reviewer for raising this important question. In the revised manuscript, we no longer describe the embedding of BC cores. Instead, following the reviewer's comment and consistent with several other related suggestions, we have thoroughly rewritten this part of the text. In the updated analysis, the ratio of $C_{sca\_measured}/C_{sca\_modeled}$ derived from SP2 measurements and core-shell Mie calculations is treated as an optical proxy that indirectly reflects the $M_R$-dependent optical transition behavior of BC-containing particles. This ratio is used to infer changes in particle compaction and coating state based solely on optical responses, rather than to identify a precise experimental $M_R$ value corresponding to full embedding. As now clarified, the transition ranges reported in the manuscript represent optically inferred transition states, not morphologically observed boundaries (Lines 346-373)

"*The SP2 measures the scattering cross-section ($C_{sca}$) of single BC-containing particles. The comparison between measured and the modeled (by Core-shell Mie model) $C_{sca}$ serves as an optical proxy of changes in BC compaction and coating state, reflecting the evolution of optical properties during aging process. Fig. 3 presents the variation of the ratio $C_{sca\_measured}/C_{sca\_modeled}$ at wavelength of 1064 nm with $M_R$ under different $M_P$. When $M_R$ is relatively low, $C_{sca\_measured}/C_{sca\_modeled}$ is less than 1, suggesting that the BC cores may exist in a fractal structure, remain bare, or are not fully embedded in the coating materials. Consequently, the measured $C_{sca}$ is lower than the $C_{sca}$ predicted by the core-shell Mie model. This observation aligns with Liu et al. (2017a), who classified such BC-containing particles as externally mixed. As $M_R$ increases, $C_{sca\_measured}/C_{sca\_modeled}$ also increases, indicating the BC particles becomes more compact and more thoroughly coated, transitioning toward a core-shell structure (Corbin et al., 2023). Following previous studies (Liu et al., 2017a; Liu et al., 2020), we describe this stage as a "transition state". In this work, the transition state is neither defined by a fixed $M_R$ threshold nor by any directly observed morphological boundary. Instead, it reflects an optically inferred state in which scattering enhancement increases markedly, with $M_R$ ranges of 1.78-6.34 (Case 1), 1.43-3.78 (Case 2), and 1.45-4.19 (Case 3). The higher $M_R$ thresholds observed in Case 2 and Case 3 indicate that under polluted conditions, BC particles can reach an optically core-shell-like state with comparatively less coating material. This likely reflected accelerated aging driven by enhanced secondary formation and condensation of inorganics and organics on BC, facilitated by stagnant meteorological conditions (low wind speed). Such conditions promote efficient coating growth on BC-containing particles, strengthening their light-absorption capability and leading to high $E_{abs}$. Therefore, compared with Case 1, BC in Case 2 and*

*Case 3 required less coating material to reach the core-shell configuration. When $M_R$ exceeds the transition state range, the ratio $C_{sca\_measured}/C_{sca\_modeled}$ becomes relatively stable, suggesting that the BC particles behave optically like compact, spherical core-shell structures."*

**[Comment 19]** Figure S1, CPAS not defined. ACSM not defined. Define all acronyms in caption.

**Response:** We appreciate the reviewer's comment. As all instrument acronyms are listed and defined in Table S1, we prefer not to repeat them in each figure caption to avoid redundancy.

**[Comment 20]** Line 323, "during the haze period... chemical reactions produced a lareg number of inorganic substances..." How do you know this?

**Response:** We thank the reviewer for the comment. As this point was also raised by **reviewer 1 [Comment 31]**, we have provided a detailed clarification in our previous response. Briefly, the relevant section has been thoroughly revised to remove unsupported statements and to base the discussion solely on our measured chemical composition data. The updated analysis shows that only secondary nitrate exhibits a consistent association with increasing $E_{abs\_measured}$, while other inorganic and organic components do not show significant effects. This revised, data-driven interpretation is now fully reflected in the revised manuscript (Lines 415-457).

*"The transitional-state particles are BC-containing particles in the process of evolving from loosely aggregated fractal-like structures toward quasi–core–shell configurations (Moffet et al., 2016; Moteki and Kondo, 2007). The abundance of transitional-state particles varies notably under different atmospheric conditions, directly influencing the measured $E_{abs}$ (Liu et al., 2017a). During clean days (Case 1), the atmospheric environment was characterized by low $PM_1$ concentrations, weak secondary formation, and highly variable coating conditions. Under such conditions, our measurements show that BC-containing particles were dominated by transitional-state structures (Fig. S9), representing the intermediate stage between externally mixed aggregates and fully developed quasi–core–shell structures. The limited and heterogeneous coating distribution on these particles substantially weakens the lensing effect, resulting in lower measured $E_{abs}$ (Peng et al., 2016). Because the core-shell Mie model inherently assumes a uniform and concentric coating, it does not accurately represent the optical behavior of these transitional particles, leading to a pronounced overestimation of measured $E_{abs}$ during Case 1. This indicates that, under clean conditions, the optical properties of transitional-state particles are the key driver of the model-observation discrepancy. In contrast, the haze period (Case 2) represents a more aged and heavily coated aerosol environment and provides a useful reference for understanding the factors influencing the measured $E_{abs}$. During Case 2, the high aerosol loading and elevated bulk-averaged $M_R$ were largely influenced by regional transport, as air masses at 100 m, 500 m, and 1000 m all followed similar pathways from the northern Yangtze River Delta into northern Zhejiang (Fig. 1c and Fig. S7). Stagnant meteorological conditions, elevated relative humidity, and enhanced oxidative capacity further facilitated vigorous liquid-phase and photochemical reactions, promoting the abundant formation of secondary coatings on BC surfaces (Peng et al., 2016). Notably, our observations show that $E_{abs}$ increases systematically with the increasing contribution of secondary nitrate (Fig. S8), consistent with the fact that nitrate-rich conditions enhance aqueous-phase oxidation and accelerate the formation of thick inorganic coatings (Liu et al., 2017b). As a result, a much larger fraction of BC-containing particles exhibited internally mixed, quasi–core-shell structures rather than transitional states (Fig. S9), which explains why the core-shell Mie model*

*performs substantially better for Case 2 than for Case 1. This contrast reinforces the central role of transitional-state particles in determining measured $E_{abs}$ when coatings are sparse, irregular, or partially developed. Given the strong influence of transitional-state particles on measured $E_{abs}$ in Case 1, precise constraints on their optical behavior are crucial for improving $E_{abs}$ estimates across different atmospheric scenarios. To address this, an empirical formula based on optical measurements was developed to estimate the $E_{abs}$ of BC-containing particles in the "transition state", derived from fitting the measured $C_{sca}$ against $M_R$ (Fig. 3d-3f). By applying this empirical formula to the calculation of $E_{abs}$, the resulting value for Case 1 was 1.21 ± 0.01. For Case 2 and Case 3, the $E_{abs}$ calculated using the same formula ($E_{abs\_param}$) remained slightly lower than the $E_{abs\_measured}$, but the deviation was within 20%, demonstrating the reliability of the approach across different atmospheric conditions."*

**[Comment 21]** Line 327, please define "embedding pattern"

**Response:** We thank the reviewer for this comment. In response to this and related reviewer suggestions, we have removed the term "embedding pattern" and substantially revised this section to avoid implying any morphology-based classification that cannot be directly observed. The revised text now focuses on the concept of transitional-state BC-containing particles, defined solely based on their $M_R$-dependent optical transition behavior inferred from SP2 measurements and Mie-model calculations. The updated description is provided in Lines 415-419 of the revised manuscript.

*"The transitional-state particles are BC-containing particles in the process of evolving from loosely aggregated fractal-like structures toward quasi–core–shell configurations (Moffet et al., 2016; Moteki and Kondo, 2007). The abundance of transitional-state particles varies notably under different atmospheric conditions, directly influencing the measured $E_{abs}$ (Liu et al., 2017a)."*

**[Comment 22]** Line 331, either the formation of BC coatings was "unfavourable" or the plume was less aged.

**Response:** We thank the reviewer for this comment. In response, and taking into account similar feedback from other reviewers, we have thoroughly revised this section to clarify the roles of coating formation conditions and plume aging. The discussion now avoids unsupported statements and provides a more data-driven interpretation (see Lines 415-457 in the revised manuscript).

*"The transitional-state particles are BC-containing particles in the process of evolving from loosely aggregated fractal-like structures toward quasi–core–shell configurations (Moffet et al., 2016; Moteki and Kondo, 2007). The abundance of transitional-state particles varies notably under different atmospheric conditions, directly influencing the measured $E_{abs}$ (Liu et al., 2017a). During clean days (Case 1), the atmospheric environment was characterized by low $PM_1$ concentrations, weak secondary formation, and highly variable coating conditions. Under such conditions, our measurements show that BC-containing particles were dominated by transitional-state structures (Fig. S9), representing the intermediate stage between externally mixed aggregates and fully developed quasi–core–shell structures. The limited and heterogeneous coating distribution on these particles substantially weakens the lensing effect, resulting in lower measured $E_{abs}$ (Peng et al., 2016). Because the core-shell Mie model inherently assumes a uniform and concentric coating, it does not accurately represent the optical behavior of these transitional particles, leading to a pronounced overestimation of measured $E_{abs}$ during Case 1. This indicates that, under clean*

*conditions, the optical properties of transitional-state particles are the key driver of the model-observation discrepancy. In contrast, the haze period (Case 2) represents a more aged and heavily coated aerosol environment and provides a useful reference for understanding the factors influencing the measured $E_{abs}$. During Case 2, the high aerosol loading and elevated bulk-averaged $M_R$ were largely influenced by regional transport, as air masses at 100 m, 500 m, and 1000 m all followed similar pathways from the northern Yangtze River Delta into northern Zhejiang (Fig. 1c and Fig. S7). Stagnant meteorological conditions, elevated relative humidity, and enhanced oxidative capacity further facilitated vigorous liquid-phase and photochemical reactions, promoting the abundant formation of secondary coatings on BC surfaces (Peng et al., 2016). Notably, our observations show that $E_{abs}$ increases systematically with the increasing contribution of secondary nitrate (Fig. S8), consistent with the fact that nitrate-rich conditions enhance aqueous-phase oxidation and accelerate the formation of thick inorganic coatings (Liu et al., 2017b). As a result, a much larger fraction of BC-containing particles exhibited internally mixed, quasi–core-shell structures rather than transitional states (Fig. S9), which explains why the core-shell Mie model performs substantially better for Case 2 than for Case 1. This contrast reinforces the central role of transitional-state particles in determining measured $E_{abs}$ when coatings are sparse, irregular, or partially developed. Given the strong influence of transitional-state particles on measured $E_{abs}$ in Case 1, precise constraints on their optical behavior are crucial for improving $E_{abs}$ estimates across different atmospheric scenarios. To address this, an empirical formula based on optical measurements was developed to estimate the $E_{abs}$ of BC-containing particles in the "transition state", derived from fitting the measured $C_{sca}$ against $M_R$ (Fig. 3d-3f). By applying this empirical formula to the calculation of $E_{abs}$, the resulting value for Case 1 was 1.21 ± 0.01. For Case 2 and Case 3, the $E_{abs}$ calculated using the same formula ($E_{abs\_param}$) remained slightly lower than the $E_{abs\_measured}$, but the deviation was within 20%, demonstrating the reliability of the approach across different atmospheric conditions.*"

**References:**

Adachi, K. and Buseck, P. R.: Changes of ns-soot mixing states and shapes in an urban area during CalNex, Journal of Geophysical Research-Atmospheres, 118, 3723-3730, 10.1002/jgrd.50321, 2013.

Bond, T. C. and Bergstrom, R. W.: Light Absorption by Carbonaceous Particles: An Investigative Review, Aerosol Science and Technology, 40, 27-67, 10.1080/02786820500421521, 2006.

Corbin, J. C., Modini, R. L., and Gysel-Beer, M.: Mechanisms of soot-aggregate restructuring and compaction, Aerosol Science and Technology, 57, 89-111, 10.1080/02786826.2022.2137385, 2023.

Gysel, M., Laborde, M., Olfert, J. S., Subramanian, R., and Gröhn, A. J.: Effective density of Aquadag and fullerene soot black carbon reference materials used for SP2 calibration, Atmospheric Measurement Techniques, 4, 2851-2858, 10.5194/amt-4-2851-2011, 2011.

Liu, D., Taylor, J. W., Young, D. E., Flynn, M. J., Coe, H., and Allan, J. D.: The effect of complex black carbon microphysics on the determination of the optical properties of brown carbon: BC morphology on BrC optical properties, Geophysical Research Letters, 42, 613-619, 10.1002/2014GL062443, 2015.

Liu, D., Allan, J. D., Young, D. E., Coe, H., Beddows, D., Fleming, Z. L., Flynn, M. J., Gallagher, M. W., Harrison, R. M., Lee, J., Prevot, A. S. H., Taylor, J. W., Yin, J., Williams, P. I., and Zotter, P.: Size distribution, mixing state and source apportionment of black carbon aerosol in London during

wintertime, Atmospheric Chemistry and Physics, 14, 10061-10084, 10.5194/acp-14-10061-2014, 2014.

Liu, D., Whitehead, J., Alfarra, M. R., Reyes-Villegas, E., Spracklen, D. V., Reddington, C. L., Kong, S., Williams, P. I., Ting, Y.-C., Haslett, S., Taylor, J. W., Flynn, M. J., Morgan, W. T., McFiggans, G., Coe, H., and Allan, J. D.: Black-carbon absorption enhancement in the atmosphere determined by particle mixing state, Nature Geoscience, 10, 184-188, 10.1038/ngeo2901, 2017a.

Liu, H., Pan, X., Liu, D., Liu, X., Chen, X., Tian, Y., Sun, Y., Fu, P., and Wang, Z.: Mixing characteristics of refractory black carbon aerosols at an urban site in Beijing, Atmospheric Chemistry and Physics, 20, 5771-5785, 10.5194/acp-20-5771-2020, 2020.

Liu, Y., Wu, Z., Wang, Y., Xiao, Y., Gu, F., Zheng, J., Tan, T., Shang, D., Wu, Y., Zeng, L., Hu, M., Bateman, A. P., and Martin, S. T.: Submicrometer Particles Are in the Liquid State during Heavy Haze Episodes in the Urban Atmosphere of Beijing, China, Environmental Science & Technology Letters, 4, 427-432, 10.1021/acs.estlett.7b00352, 2017b.

Moffet, R. C., O'Brien, R. E., Alpert, P. A., Kelly, S. T., Pham, D. Q., Gilles, M. K., Knopf, D. A., Laskin, A., Pacific Northwest National Lab, R. W. A. E. M. S. L., Lawrence Berkeley National Lab, B. C. A., and Stony Brook Univ, S. B. N. Y.: Morphology and mixing of black carbon particles collected in central California during the CARES field study, Atmospheric chemistry and physics, 16, 14515-14525, 10.5194/acp-16-14515-2016, 2016.

Moteki, N. and Kondo, Y.: Effects of Mixing State on Black Carbon Measurements by Laser-Induced Incandescence, Aerosol Science and Technology, 41, 398-417, 10.1080/02786820701199728, 2007.

Naseri, A., Corbin, J. C., and Olfert, J. S.: Comparison of the LEO and CPMA-SP2 techniques for black-carbon mixing-state measurements, Atmos. Meas. Tech., 17, 3719-3738, 10.5194/amt-17-3719-2024, 2024.

Peng, J., Hu, M., Guo, S., Du, Z., Zheng, J., Shang, D., Zamora, M. L., Zeng, L., Shao, M., Wu, Y.-S., Zheng, J., Wang, Y., Glen, C. R., Collins, D. R., Molina, M. J., and Zhang, R.: Markedly enhanced absorption and direct radiative forcing of black carbon under polluted urban environments, Proceedings of the National Academy of Sciences, 113, 4266-4271, 10.1073/pnas.1602310113, 2016.

Radney, J. G., You, R. A., Ma, X. F., Conny, J. M., Zachariah, M. R., Hodges, J. T., and Zangmeister, C. D.: Dependence of Soot Optical Properties on Particle Morphology: Measurements and Model Comparisons, Environmental Science & Technology, 48, 3169-3176, 10.1021/es4041804, 2014.

Wang, T. T., Zhao, G., Tan, T. Y., Yu, Y., Tang, R. Z., Dong, H. B., Chen, S. Y., Li, X., Lu, K. D., Zeng, L. M., Gao, Y. Q., Wang, H. L., Lou, S. R., Liu, D. T., Hu, M., Zhao, C. S., and Guo, S.: Effects of biomass burning and photochemical oxidation on the black carbon mixing state and light absorption in summer season, Atmospheric Environment, 248, 10.1016/j.atmosenv.2021.118230, 2021.

Zanatta, M., Gysel, N., Bukowiecki, T., Müller, E., and Weingartner: A European aerosol phenomenology-5: Climatology of black carbon optical properties at 9 regional background sites across Europe, Atmospheric Environment, 2016.

Zhang, R. Y., Khalizov, A. F., Pagels, J., Zhang, D., Xue, H. X., and McMurry, P. H.: Variability in morphology, hygroscopicity, and optical properties of soot aerosols during atmospheric processing, Proc. Natl. Acad. Sci. U. S. A., 105, 10291-10296, 10.1073/pnas.0804860105, 2008.

Zhang, Y., Zhang, Q., Cheng, Y., Su, H., Li, H., Li, M., Zhang, X., Ding, A., and He, K.: Amplification of light absorption of black carbon associated with air pollution, Atmospheric Chemistry and Physics, 18, 9879-9896, 10.5194/acp-18-9879-2018, 2018.

Zhao, G., Shen, C., and Zhao, C.: Technical note: Mismeasurement of the core-shell structure of black carbon-containing ambient aerosols by SP2 measurements, Atmospheric Environment, 243, 117885, 10.1016/j.atmosenv.2020.117885, 2020.

---

## Author Comment (AC3)

**Response to Reviewer #3:**

The manuscript treats an important aspect of black carbon optical properties with innovative techniques. Overall, the topic and novelty fulfil the requirements for ACP publication. Some work still to be done improve the readability of the manuscript and especially ensure that robustness of the measured properties. The article requires consistent modification.

**[Comment 1] NOMENCLATURE AND READABILITY**: The use of abbreviations and symbols requires careful attention. Several parameters (particularly those distinguishing measured from modeled quantities) are difficult to follow, which hinders comprehension, especially in the results section. I strongly recommend introducing a summary table listing each property, its abbreviation, and whether it refers to a measurement or a modeled value. Improving consistency here would greatly enhance clarity for readers unfamiliar with the measurement framework.

**Response:** We appreciate the reviewer's suggestion regarding the clarity of abbreviations and symbols. Following this recommendation, we have now compiled a comprehensive summary table that lists all parameters, their abbreviations, and whether they refer to measured or modeled quantities. This table has been added as Table S1 in the revised Supplementary Information. We believe this addition substantially improves the readability and consistency of the manuscript, particularly for readers who are less familiar with the measurement framework.

*Table S1. The abbreviation of this study*

| Abbreviation | Full Name |
|---|---|
| BC | Black carbon |
| $E_{abs}$ | light absorption enhancement |
| $M_R$ | Coating-to-core mass ratio of single BC-containing particle |
| DRF | direct radiative forcing |
| bulk-averaged $M_R$ | Averaged $M_R$ of bulk BC-containing particle |
| $C_{sca}$ | the scattering cross-section |
| $C_{sca\_measured}$ | the measured scattering cross-section |
| $C_{sca\_modeled}$ | the modeled scattering cross-section by core-shell Mie model |
| $M_p$ | the mass of a BC-containing particle |
| $M_{BC\_core}$ | the mass of BC core |
| MAC | the mass absorption cross section |
| $E_{abs\_measured}$ | the measured light absorption enhancement |
| $MAC_{BC\_core\_measured}$ | the MAC of uncoated BC particles extrapolated from measured MAC values of BC-containing particles |
| $D_c$ | the BC core size |
| $D_p/D_c$ | BC coating thickness |
| RI | the refractive index |
| $MAC_{BC\_coated\_modeled}$ | modeled MAC of coated BC |
| WS | wind speed |
| RH | relative humidity |
| $E_{abs\_uniform}$ | the $E_{abs}$ calculated using the traditional core-shell Mie model |
| $\Delta E_{abs}$ | $E_{abs\_uniform}$-$E_{abs\_measured}$ |
| $E_{abs\_resolved}$ | the $E_{abs}$ calculated from each BC-containing particle using the core-shell Mie |

|  |  |
|---|---|
| | *model* |
| $E_{abs\_param}$ | *the $E_{abs}$ obtained based on the empirical scheme developed in this study* |
| *CPMA* | *Centrifugal Particle Mass Analyzer* |
| *SP2* | *single-particle soot photometer* |
| *PSL* | *polystyrene latex spheres* |
| *TOF-ACSM X* | *time-of-flight aerosol chemical speciation monitor* |
| *CAPS-ALB* | *Multi-Wavelength Cavity Attenuated Phase Shift Single-Scattering Albedo Monitor* |
| *DMA* | *Differential Mobility Analyzer* |
| *LEO* | *the leading-edge-only technique* |

**[Comment 2] INSTRUMENTAL DESCRIPTION AND UNCERTAINTIES:** The methodology section currently lacks sufficient citations and discussion of measurement uncertainties. Given that the study combines multiple instruments in a novel configuration, these details are crucial. The uncertainties of the CPMA–SP2 tandem system, as well as of derived quantities such as $M_p$ and M_BC, should be clearly stated and discussed in the context of previous literature. Since many key quantities in the analysis are expressed as ratios, unquantified uncertainties may propagate and influence the reported variability. In line with other reviewers' remarks, I encourage the authors to describe the CPMA–SP2 system and data processing steps in greater detail, including how calibration and error propagation were handled.

**Response:** We appreciate the reviewer's insightful comment. In the revised manuscript, we have substantially expanded the methodological description of the CPMA-SP2 system and clarified all associated measurement uncertainties. Specifically, we now report the uncertainties of $M_p$ (5%), $M_{BC\_core}$ (10%), single-particle $M_R$ (~11%), and bulk-averaged $M_R$ (~7%), and the absorption coefficient (15-20%), as well as the propagated uncertainties for $MAC_{BC\_coated\_measured}$ (18-22%), $MAC_{BC\_core\_measured}$ (19-23%), and $E_{abs\_measured}$ (26-32%). The corresponding uncertainty-estimation procedures and calibration details have been added to Text S2, and the resulting values are clearly stated at the relevant locations in the revised manuscript (Lines 98-225). Some key description can be found as follows:

Lines 111-112 ($M_p$): *"According to the instrument manual, the mass accuracy of the CPMA is approximately 5%."*

Lines 114-116 ($M_{BC\_core}$): *"……., and the uncertainty associated with the SP2-derived BC core mass ($M_{BC\_core}$) is approximately 10% (Laborde et al., 2012)."*

Lines 152-155 (the absorption coefficient): *"Absorption was calculated as the difference between extinction and scattering, with estimated uncertainties of ~1-10% for both extinction and scattering (Modini et al., 2021), leading to a conservative absorption uncertainty of ~15-20% for the submicron BC particles considered."*

Lines 180-181 (single-particle $M_R$): *"Considering the uncertainties of $M_p$ (5%) and $M_{BC\_core}$ (10%), the uncertainty of $M_R$ for a single BC-containing particle was approximately 11% (Text S2)."*

Lines 185-187 (bulk-averaged $M_R$): *"Propagating these uncertainties to the hourly mass-weighted calculation resulted in an uncertainty of approximately 7% for the bulk-averaged $M_R$ (Text S2)."*

Lines 221-224 ($MAC_{BC\_core\_measured}$ and $E_{abs\_measured}$): *"Based on our error propagation analysis,*

*which accounts for measurement uncertainties in particle absorption and BC mass as well as the standard error of the extrapolation, the estimated uncertainty of $MAC_{BC\_core\_measured}$ is approximately 19-23% (Text S2). And the uncertainty of $E_{abs\_measured}$ is approximately 26 - 32% (Text S2)."*

I hope this revision improves the clarity for readers.

**[Comment 3] RESULTS AND INTERPRETATION:** The results section shows promising potential for impact, but several points require clarification. The authors could strengthen the manuscript by expanding the discussion on how the proposed "transition-state" correction scheme might be generalized to other atmospheric environments. For example, conditions in rural or biomass-burning regions, or during other seasons, may produce distinct coating compositions and morphology evolution pathways. Explaining how the correction parameters (e.g., MR thresholds or morphology indicators) could adapt to such conditions would enhance the broader applicability of the method. The method for defining the three "cases" should also be revisited. It appears that only Case 2 represents a specific or anomalous event compared to the rest of the campaign. I suggest first describing the overall meteorological and bulk aerosol conditions and then examining how the optical properties vary under those regimes. This would provide a more physically grounded interpretation of the case classification. At present, the combined issues of unclear nomenclature and insufficient methodological detail reduce the understanding of the results, being the ultimate limiting factor of the manuscript.

**Response:** We thank the reviewer for these constructive comments. Following this suggestion-and in conjunction with the feedback from the other reviewers—we have substantially revised the Results section. Specifically, we have: (1) expanded the discussion on the potential generalization of the transition-state correction scheme to other atmospheric environments (e.g., rural, biomass-burning, and different seasonal conditions); (2) clarified how key parameters such as $M_R$ thresholds of transition state may adapt under varying coating conditions; and (3) reorganized the case definitions by first presenting the overall meteorological and bulk aerosol characteristics before examining the case-dependent optical behavior. We believe these revisions effectively address the reviewer's concerns and greatly improve the readability and interpretability of the Results section. Please refer to the revised manuscript (Section 3.1-3.3).

**SPECIFIC COMMENTS**
**[Comment 4]** L1: title. I am not convinced by "heterogeneity". What does it mean in this context? It is a very general term that, alone, does not convey a unanimous message.

**Response:** We thank the reviewer for this comment. In our study, "heterogeneity" specifically refers to the variability in the mass ratio of coatings to BC cores among particles. To clarify this and in line with the suggestions of Reviewer 2 (Comment 3), we have revised the title to:

"*Effects of Mass Ratio Heterogeneity and Coating-Related Optical Characteristics on the Light Absorption Enhancement of Black Carbon-Containing Particles.*"

**[Comment 5]** L35-38: Eabs is not contextualized, not properly described. Enhancement with respect to? Please provide a short description.

**Response:** We thank the reviewer for pointing this out. In the revised manuscript, we have clarified the definition of BC $E_{abs}$ in the context of particle coatings. Specifically, $E_{abs}$ is defined as the ratio

of absorption by coated BC particles to that of uncoated BC cores, capturing the increase in absorption due to the presence of coatings. This definition is now included in the revised manuscript (Lines 35-41):

"*According to IPCC assessments, the global effective DRF of BC ranges from $-0.28$ to $0.41$ $W/m^2$ (Szopa et al., 2021). To represent the effect of BC particle coatings on absorption, most climate models (Bauer et al., 2013; Chen et al., 2024; Stier et al., 2005; Wang et al., 2023; Zhang et al., 2025b) estimate BC light absorption enhancement ($E_{abs}$) using Mie theory, defined as the ratio of absorption by coated BC to that of uncoated BC cores, under the assumption of a uniform core-shell structure where BC core is fully coated by coating materials.*"

**[Comment 6]** L51: what do you mean with heterogeneity?

**Response:** We thank the reviewer for the comment. In this study, "heterogeneity" specifically refers to the variability or dispersion in the mass ratio ($M_R$) of the coating to BC cores among individual particles. This definition clarifies that we focus on the distribution of core-shell mass ratios. Notably, several previous studies have also used the term "heterogeneity" to characterize similar variability in BC particle properties (Fierce et al., 2020; Fierce et al., 2016; Zeng et al., 2024), supporting our usage of the term in this context.

**[Comment 7]** L67: please avoid the use of "fortunately", it undermines the preparation and thoughts behind your research. Leaving the reason of the positive outcome of your work to luck.

**Response:** We thank the reviewer for the comment. The word "fortunately" has been removed from the manuscript, as suggested. The revised sentence can be found at lines 88-89:

"*Field measurements revealed the coexistence of high, medium, and low $E_{abs}$ under high bulk-averaged $M_R$ conditions.*"

**[Comment 8]** L80-84. If available, I suggest adding 1 or 2 references describing the site and its representativity.

**Response:** We thank the reviewer for the suggestion. References describing the Hangzhou site and its representativeness have been added (Zhang et al., 2025a; Qian et al., 2025). These are previous studies from our group, conducted at the same observation location, further supporting the representativity (Lines 83-85).

"*In this study, a suite of state-of-the-art instruments were employed to simultaneously capture the magnitude and temporal of BC $M_R$ in Hangzhou, China (Zhang et al., 2025a; Qian et al., 2025).*"

**[Comment 9]** L86-109. I suggest describing a bit more each instrument alone. With the use of references, which is limited. Here some old works about SP2 describing its principle: (Stephens et al., 2003; Moteki and Kondo, 2010) and calibrations: (Gysel et al., 2011). I suggest a recent paper exploring the operational limits of the SP2 (Schwarz et al., 2022). This is the major reference of the CPMA: (Olfert and Collings, 2005). For the tandem combination of SP2 with mass analysers I suggest a relatively old review (Cross et al., 2010) and a more recent set of papers (Liu et al., 2022; Naseri et al., 2022; Zanatta et al., 2025). DMT 2011. This is an odd reference. The ACQUADAG scaling factor to fullerene should be slightly better accounted for. DMT 2011 is and odd reference. The original reference should be (Baumgardner et al., 2012; Laborde et al., 2012). Please do the

same for the ACSM. The CAPS-SSA is fully detailed by (Modini et al., 2021). Overall, these works report all the error associated with the single measurements, which will propagate substantially for the application intended in the current manuscript. None of these are described.

**Response:** We thank the reviewer for the suggestions regarding instrument descriptions and relevant references. All suggested references have now been added. Regarding DMT 2011, this refers to the Aquadag aerosols supplier and production year rather than a literature reference; its previous placement led to confusion, which has now been clarified. Additionally, we have provided concise descriptions of each instrument and performed a rough assessment of the propagated errors associated with individual measurements, as relevant for the applications in revised manuscript (see **[Comment 2]**).

**[Comment 10]** L93: I would expect the SP2 showing the multi charged particles. How exactly was the mass MBC calculated?

**Response:** We thank the reviewer for the comment. The treatment of multiply charged particles has already been detailed in Fig. S10 and Fig. S11 and the accompanying Supplementary Text S1. The BC core mass ($M_{BC}$) was calculated based on the linear relationship between the SP2 incandescence peak height and BC mass established during calibration. Relevant description has been added to the revised manuscript (Lines 135-141).

*"In the subsequent data processing, measurements from the CPMA-SP2 system were first corrected for multiple charging effects. Additional corrections, including transfer function, detection efficiency, and time delay, were also applied, as detailed in the Text S1. The mass of each BC core ($M_{BC\_core}$) was then calculated from the SP2 incandescence signal using the calibration described above, with a correction factor of 0.75 applied to the peak height (Liu et al., 2020; Liu et al., 2014; Zhang et al., 2018; Gysel et al., 2011)."*

**[Comment 11]** L115. What model and company? Please be consistent with previous notation.

**Response:** We have added the model and company information for the Nephelometer to be consistent with previous notation (Aurora 3000, Acoem; see Line 161).

**[Comment 12]** L122: CAPS and nephelometer may well respond to PSL, especially small PSL. Truncation error may become more and more important with larger particles and especially irregular particles…such as ramified fresh BC. Please provide a small statement about it. Was truncation corrected?

**Response:** We thank the reviewer for the comment regarding truncation error. As reported in Madini et al. (2021), the CAPS PMssa instrument can measure extinction ($\sigma_{ext}$) and scattering ($\sigma_{sca}$) coefficients with high accuracy, with errors on the order of 1-10%. However, absorption derived via the extinction-minus-scattering method is subject to substantial subtractive error amplification, particularly for highly scattering or irregular particles such as ramified fresh BC. The main source of this uncertainty is the truncation of near-forward and near-backward scattered light that is not captured by the instrument. Using standard error propagation for subtraction, the resulting uncertainty in $\sigma_{abs}$ can be estimated as $\sqrt{\sigma_{ext}^2 + \sigma_{sca}^2}$. For submicron BC particles in this study, this leads to a conservative estimate of absorption uncertainty on the order of 15-20%, accounting for both instrument precision and potential truncation effects. We did not apply explicit truncation

corrections, because our focus is on the relative enhancement of absorption with increasing $M_R$ rather than absolute absorption values. Besides, the nephelometer measurements in this study were used solely to compare with CAPS results and assess the stability and reliability of the CAPS measurements, and no truncation correction was applied. The presence of CAPS measurement uncertainties and other contributing factors to absorption uncertainty have been noted in the Methods section (lines 149-157).

*"The aerosol extinction and scattering coefficient (Fig. S3) at wavelength of 440, 530 and 630 nm were measured by Multi-Wavelength Cavity Attenuated Phase Shift Single-Scattering Albedo Monitor (CAPS-ALB, Shoreline) (Weber et al., 2022). Absorption was calculated as the difference between extinction and scattering, with estimated uncertainties of ~1-10% for both extinction and scattering (Modini et al., 2021), leading to a conservative absorption uncertainty of ~15-20% for the submicron BC particles considered. No explicit truncation correction was applied, as the analysis focuses on the relative enhancement of absorption with $M_R$ rather than absolute values."*

**[Comment 13]** L127: Well, we "assume" that everything is working properly. This is why is important to estimate, even roughly, the uncertainty.

**Response:** Thank you for your comment. We agree that estimating the uncertainty is important. We have performed a rough evaluation of the uncertainties associated with the parameters discussed in the manuscript. The details of this assessment are provided in **[Comment 2].**

**[Comment 14]** L132-133: I have a couple of questions here. The CPMA is capable of selecting particles based on their mass to charge ratio. Hence, it is recommended, even by the manufacturer, to run the CPMA after a neutralizer/charger. From the schematics of Figure S1 it looks like there was no neutralizer. What it is the additional uncertainty of running the CMPA-SP2 setup without a charger? Single and multi-charged peaks, should be visible in the mass distribution provided by the SP2. I wonder if the authors quantified the single particle BC mass (MBC, by fitting the SP2 mass distribution, including only the first peak (single charge) or fitting the full distribution). This technical detail may influence all the result sections. Hence need to be fully and properly described. Regarding the sampling collection. Unfortunately, Figure S2 shows that the counting efficiency of the SP2 is far from 100%, especially below 100 nm (Figure S2a). It is also surprising that the SP2 counts systematically more than the CPC. I presume that some setting in the SP2 were not properly configured or that the CPC had some counting issues. Moreover, why the two incandescence detectors should have a different (linear and non-linear) mass/incandescence relationship?

**Response:**

**1. Regarding the CPMA-SP2 multiple charge issue:** We would like to clarify that the influence of multiply charged particles ($q > 1$) has been explicitly considered in our CPMA-SP2 measurements. An X-ray aerosol neutralizer (TSI 3088) was installed upstream of the CPMA to charge the aerosols before entering the classifier (Lines 121-123). Although we did not directly use the inversion code from Liu et al. (2017) or Naseri et al. (2024), we have applied an independent multiple charge correction procedure to the data, allowing us to identify and select only singly charged particles. All relevant details had already been included in Text S1 and Figure S10 and Figure S11. To facilitate readers' understanding of the data correction procedures and the CPMA-SP2 system configuration,

we have added the relevant descriptions in the Methods (Lines 135-138) and Text S1 and included the aerosol neutralizer in the schematic diagram of Figure S1.

**2. Regarding CPC and SP2 particle counting:** We note that the SP2 and CPC instruments have inherently different counting principles, which can lead to systematic differences in particle counts. The SP2 measures single-particle BC core properties via incandescence, whereas the CPC detects total particle number via condensation, which explains why SP2 counts can sometimes exceed CPC counts under certain conditions. Additionally, the SP2 has reduced detection efficiency for small particles, particularly below ~100 nm, as shown in Figure S2a. To ensure data reliability, our analysis was restricted to particles with BC core diameters larger than 80 nm, where the SP2 counting efficiency is sufficiently high.

**3. why the two incandescence detectors have a different (linear and non-linear) mass/incandescence relationship**: The two incandescence detectors of the SP2 exhibit different mass-signal behaviors due to their gain settings and the underlying physics. The low-gain detector operates within the linear dynamic range of the photomultiplier, so the incandescence signal increases linearly with BC mass. In contrast, the high-gain detector is designed to enhance sensitivity for small particles, but for larger BC masses the PMT enters a non-linear response regime, causing the signal to deviate from linearity. This difference in detector behavior is intrinsic to the SP2 design and explains why the mass-incandescence relationship appears linear for one detector and non-linear for the other.

Lines 121-123: "*In this setup, particles with known mass ($M_p$) selected by CPMA were injected into the SP2. An X-ray aerosol neutralizer (TSI 3088) was installed upstream of the CPMA to charge the aerosols to a Boltzmann equilibrium before entering the classifier.*"

Lines 135-138: "*In the subsequent data processing, measurements from the CPMA-SP2 system were first corrected for multiple charging effects. Additional corrections, including transfer function, detection efficiency, and time delay, were also applied, as detailed in the Text S1.*"

**[Comment 15]** L140, why distorted. It is attenuated due to the evaporation of absorbing-refractory material.

**Response:** We thank the reviewer for pointing this out. The scattering signal measured by the SP2 can indeed appear distorted due to partial evaporation of absorbing-refractory material as BC-containing particles pass through the laser beam. To address this, we have clarified the text and revised the sentence as follows (Lines 177-202):

"*The measured scattering cross section ($C_{sca\_measured}$) was obtained from the SP2 using the leading-edge-only (LEO) technique, which reconstructs the scattering signal as BC-containing particles pass through the SP2 laser beam due to partial evaporation of refractory-absorbing material. The validity of this reconstruction relies on the assumption that the leading-edge data used for fitting represents an unperturbed particle, as extensively reported in previous studies.*"

**[Comment 16]** L142-146: the LEO-fit relies on many assumptions, as correctly stated by the authors. It would be nice if they could elaborate, shortly, about the reason behind these choices. Moreover, with a similar number of assumptions (density of coating and BC cores), the optical coating thickness could be derived directly from the Mp and Mbc was this performed? Are the results coherent?

**Response:** We thank the reviewer for the comment. The validity of the LEO-fit method indeed relies on the assumption that the leading-edge data used for fitting represents an unperturbed particle. In our study, the LEO reconstruction was used solely to obtain the scattering signal of BC-containing particles, which is necessary for multi-charged particle distribution statistics and the corresponding multi-charge corrections. For the coating thickness, we directly calculated it from the CPMA-selected particle mass ($M_p$) and the BC core mass ($M_{BC}$). Previous study have demonstrated that this approach provides higher accuracy and reliability compared with estimating coating thickness based on LEO-fit reconstruction (Naseri et al., 2024). Therefore, the LEO-fit was not used for coating thickness calculations.

Besides, we have now added a description of the LEO-fit assumptions to the main revised manuscript (Lines 197-205).

*"The measured scattering cross section ($C_{sca\_measured}$) was obtained from the SP2 using the leading-edge-only (LEO) technique, which reconstructs the scattering signal as BC-containing particles pass through the SP2 laser beam due to partial evaporation of refractory-absorbing material. The validity of this reconstruction relies on the assumption that the leading-edge data used for fitting represents an unperturbed particle, as extensively reported in previous studies (Liu et al., 2014; Zhang et al., 2016; Brooks et al., 2019; Gao et al., 2007; Zhang et al., 2020). Note only particles with successfully fitted LEO signals are considered in the optical property calculations."*

**[Comment 17]** L149: define the MAC.

**Response:** We thank the reviewer for the comment. We have clarified the definition of the mass absorption cross section (MAC) in the revised manuscript (Lines 212-215).

*"The light absorption enhancement of BC-containing particles is defined as the ratio of the mass absorption cross section (MAC) of the coated and uncoated BC-containing particles (Eq. 3), Here, MAC is defined as the particle light absorption cross section normalized by the BC mass, representing the light absorption per unit mass of BC."*

**[Comment 18]** L154: I like the approach of deriving the MAC of uncoated BC using this extrapolation. However, this MAC (no units) for uncoated BC results to be slightly higher than previous estimations in European urban (Savadkoohi et al., 2024) rural (Zanatta et al., 2016) and the canonical 7.5 m/ g of (Bond and Bergstrom, 2006). This could also be due to the high variability of MAC itself across sites and seasons, but also to strong uncertainties related with MAC (absorption and BC mass) and bulk MR (width of the distribution of particles exiting the CPMA and method to quantify the single particle mass with the SP2). Overall, I notice a lack in providing context to these findings and assumptions. The MACBC_core is fundamental to all the results presented in the paper, hence, even small errors may substantially modify the quantification of the enhancement. The authors must provide more details on their methods and uncertainties, and put all of these consideration with the context of recent literature.

**Response:** We thank the reviewers for their valuable comments. The reported $MAC_{BC\_core\_measured}$ of 9.08 $m^2$/g at 630 nm corresponds to uncoated BC, representing the intrinsic light absorption efficiency of BC without influence from any coating materials. This value is independent of the mixing state and reflects the inherent optical property of BC itself. The slightly higher value compared to 7.5 $m^2$/g reported by Bond and Bergstrom (2006), can be attributed to several plausible

factors, including variations in BC morphology (aggregate compactness, fractal dimension, and cluster structure), instrumental and calibration effects (SP2 calibration with Aquadag standards, detector response, and limited sampling statistics), and site-specific environmental variability (differences in BC sources and seasonal atmospheric conditions).

Regarding uncertainties, we conducted a quantitative error propagation analysis. Considering relative uncertainties of 15-20% for absorption, 10% for SP2-measured BC mass, and the extrapolation standard error of 0.54 $m^2$ $g^{-1}$, the $MAC_{BC\_core\_measured}$ uncertainty is estimated to be ~19% - 23%. Consequently, the $E_{abs\_measured}$ has a propagated uncertainty of ~26% - 32% (Text S2). This analysis demonstrates that, while the $MAC_{BC\_core\_measured}$ value is slightly higher than the commonly cited value of 7.5 $m^2$ $g^{-1}$ (Bond and Bergstrom, 2006), it remains well constrained and representative of the local atmospheric BC properties.

We have added this discussion in the revised manuscript (Lines 217-228), providing details on methods, uncertainties, and comparisons with recent literature to give proper context and highlight the robustness of our findings (Text S2).

*"The value of $MAC_{BC\_core\_measured}$ was obtained by extrapolating $MAC_{BC\_coated\_measured}$ to the limit of bulk-averaged $M_R = 0$ using linear regression. The $MAC_{BC\_core\_measured}$ at wavelength of 630 nm was 9.08 ± 0.53 $m^2$ $g^{-1}$ (mean ± 90% confidence Interval) (Fig. S6). Based on our error propagation analysis, which accounts for measurement uncertainties in particle absorption and BC mass as well as the standard error of the extrapolation, the estimated uncertainty of $MAC_{BC\_core\_measured}$ is approximately 19-23% (Text S2). And the uncertainty of $E_{abs\_measured}$ is approximately 26 - 32% (Text S2). For comparison, the $MAC_{BC\_core\_measured}$ is slightly higher than the value of 7.5 $m^2$ $g^{-1}$ recommended by Bond and Bergstrom (2006) but still within the range reported by other study (~6.5 - 17 $m^2$ $g^{-1}$) (Zanatta et al., 2016), likely due to variations in measurement methods, and site-specific atmospheric conditions."*

**[Comment 19]** L169-175: I am genuinely confused on how the MACbc introduced in equation 3 was calculated. Use a logic order when presenting variables. Is the MAC of equation 4 the same presented in equation 3?

**Response:** We thank the reviewer for the comment. We apologize for the confusion caused by the previous equation. The corrected equation has now been included in the revised manuscript. The formula has been revised as follows:

$$E_{abs\_measured} = \frac{MAC_{BC\_coated\_measured}}{MAC_{BC\_core\_measured}} \qquad (3)$$

$$MAC_{BC\_coated\_modeled} = \frac{\sum_i MAC_{BC\_coated\_modeled,i} \times M_{BC\_core,i}}{\sum_i M_{BC\_core,i}} \qquad (4)$$

**[Comment 20]** L174: The nomenclature is a bit confusing here. MACBC-core of line 174 is the same used int Equation 3 ? Or the MA_core, presented in Line 154 is used in equation 3? So, the "Mie MACBC_core" is it similar to 9.08 m2 /g. This aspect is extremely important and influences with a different weight MACMie and MAC observed with a different weight, especially in figure 2d where the delta enhancement is presented.

**Response:** We thank the reviewer for pointing out the confusion in the nomenclature. We fully agree

that the inconsistent notation could affect the interpretation of Equation 3. In the revised manuscript, we have thoroughly standardized and clarified all relevant symbols. The updated nomenclature and corresponding explanations are now explicitly provided in the Methods 2.2 and 2.3.

**[Comment 21]** L188: At what wavelength are these values provided. Could the authors state something about the absorption enhancement at different wavelengths? The high presence of organic material may change the Eabs at lower wvalenght?

**Response:** We thank the reviewer for this insightful comment. All absorption-related parameters and the subsequent analysis of $E_{abs}$ in our study are based on the optical measurements at 630 nm. This clarification has been explicitly stated in the revised manuscript. (Lines 157-159 and Lines 261-265)

Lines 157-159: "*In this study, only the measurements at 630 nm were used for subsequent analysis, as this wavelength is minimally affected by brown carbon absorption.*"

Lines 261-265: "*The average $E_{abs\_measured}$ during the sampling period in Hangzhou is 1.28 ± 0.02 (mean ± 90% confidence Interval, the same below) at wavelength of 630 nm, and the bulk-averaged $M_R$ is 3.32 ± 0.06, with average $E_{abs\_measured}$ values of 1.09 ± 0.02, 1.84 ± 0.07, and 1.55 ± 0.04 for Case 1, Case 2, and Case 3, respectively.*"

**[Comment 22]** L220-221: I recommend caution when mentioning morphology, especially in a section title. This scattering cross-section ratio is a far approximation for morphology assessment. It may be a proxy, but nothing more and must be confirm by real morphology observations such as microscopy fractal dimension or, at least DMA/CPMA density/fractal exponent measurements.

**Response:** We thank the reviewer for the helpful comment. A similar concern was also raised by Reviewer 2 (Comment 3). In response, we have removed the morphology-related descriptions in both the Abstract and the main text. These statements have been revised to explicitly reflect that our correction scheme is based on the $M_R$-dependent optical transition behavior, rather than particle morphology. We believe these changes improve the accuracy and clarity of the revised manuscript. Please see the revised manuscript.

**[Comment 23]** L222-223: what is the meaning of the sentence?

**Response:** We thank the reviewer for this comment. The sentence serves as a summary and overview for this subsection, providing a general introduction to the subsequent descriptions. To improve clarity and readability, we have revised the sentence in the manuscript as follows (Lines 315-318):

"*The coating-to-BC mass ratio ($M_R$) and the ratio of measured to modeled scattering cross sections were used to quantify the mixing state and associated optical transitions behavior of BC-containing particles, with $M_R$ serving as an important indicator of BC aging.*"

**[Comment 24]** L239: The authors states that the difference between observed and modelled enhancement depends on the variability of the standard deviation of MR. First why the log10(Mr) was used ? Second, I all honesty, it is difficult to observe any sort of correlation in the scatterplot presented in figure 2d. Especially considering that correlation coefficient and slope changes substantially among the periods. In my opinion, Figure 2s does not support the claims of the authors.

**Response:** We appreciate the reviewer's insightful question.

**1. To why use log₁₀(M_R):** The logarithmic transformation of $M_R$ ($\log_{10}(M_R)$) was applied to reduce the strong skewness in the $M_R$ distribution, which typically spans several orders of magnitude. Using $\log_{10}(M_R)$ allows the data to approximate a normal distribution, making the standard deviation a more representative measure of heterogeneity. This approach is also consistent with previous studies (Fierce et al., 2020; Huang et al., 2024).

**2. To Figure 2:** We appreciate the reviewer's careful evaluation of Fig. 2d. We would like to clarify that the purpose of this figure was not to demonstrate a statistically strong linear correlation. Instead, the figure was intended to illustrate the overall trend that the $\Delta E_{abs}$ ($E_{abs\_uniform}$-$E_{abs\_measured}$) increases with increasing SD when considering all cases together. The previously shown regression line was only for visual guidance and may have unintentionally implied a stronger correlation.

Importantly, our result is consistent with earlier studies that used mixing-state entropy to characterize BC coating diversity (Zhao et al., 2021; Riemer et al., 2019; Zeng et al., 2024). These studies reported that the model-measurement deviation in $E_{abs}$ decreases as the mixing-state entropy approaches 1, meaning that larger variability in BC coating (higher heterogeneity) leads to larger bias between modeled and measured $E_{abs}$. This is in good agreement with the trend observed in our analysis.

To avoid any potential misunderstanding, we have removed the regression equations from Fig. 2d and revised the corresponding text in the manuscript to clarify this point (Lines 327-337). We hope this revision more accurately reflects our intended message.

*"The results showed that the SD of Case 1 (0.63 ± 0.004) was greater than that of Case 3 (SD = 0.52 ± 0.012), followed by Case 2 (SD = 0.48 ± 0.005). In contrast, the $E_{abs\_measured}$ exhibited an opposite trend, suggesting that greater $M_R$ heterogeneity of BC-containing particles leads to a lower $E_{abs\_measured}$. As shown in Fig. 2d, the discrepancy between the modeled (uniform core-shell Mie model) and the measured $E_{abs}$ increases with SD, with this trend being most pronounced in Case 1, where $M_R$ heterogeneity is highest. This suggests that greater $M_R$ heterogeneity may lead to larger deviations from the uniform core-shell assumption, thereby increasing the mismatch between the modeled and measured $E_{abs}$. Such discrepancies likely due to the uniform core-shell model's simplified treatment of $M_R$ heterogeneity in BC (Romshoo et al., 2024; Wang et al., 2021)."*

**[Comment 25]** L254: please provide the wavelength

**Response:** We thank the reviewer for the careful examination. The wavelength has now been added in the revised manuscript. Specifically, the measurement was conducted at 1064 nm, and this information has been included in Line 349-351 accordingly.

*"Fig. 3 presents the variation of the ratio $C_{sca\_measured}/C_{sca\_modeled}$ at wavelength of 1064 nm with $M_R$ under different $M_p$."*

**[Comment 26]** L252-276: although the results shown in figure 3 are interesting, this section is very confusing. It is hard to understand what causes the decrease in the transition regime. Try to restructure your though in a more logic process. Could this "transition state" represent the compaction due to coating formation. This sort of natural process will reduce the optical and geometrical cross section of the particles. It is usually observed in chamber studies (e.g. Schnaiter et al., 2005; Zanatta et al., 2025) and rarely, up to my knowledge, observed in ambient conditions

(Bhandari et al., 2019). This process description could be developed further.

**Response:** We thank the reviewer for this valuable comment. We have clarified in the revised manuscript that the observed "transition state" reflects the BC aging process. During clean periods, limited secondary formation and low coating material mean that BC particles require more mass to reach an optically core-shell-like state, resulting in higher $M_R$ values for transitional-state particles. In contrast, during polluted periods, enhanced secondary formation produces sufficient coatings, so BC reaches the core-shell configuration at lower $M_R$. These differences in coating availability and atmospheric processing allow us to infer the progression of BC aging from optical observations, even though direct morphological compaction is not measured. Note while chamber studies (Zanatta et al., 2025; Schnaiter et al., 2005) directly observe coating-induced compaction, our study demonstrates that field-based optical transitions provide insight into particle aging under ambient conditions. These revisions improve the physical interpretation of the results. Incorporating the comments from other reviewers, this section has been substantially revised; see Section 3.2 and 3.3 for details.

**[Comment 27]** Section 3.3 soffers a similar issue with readability. I am not fully convinced by the reasoning behind the period separation. Only period 2 looks different from the others.

**Response:** We thank the reviewer for this comment. In response, and based on feedback from you and the other reviewers, we have substantially revised Section 3.3 to improve readability and clarify the reasoning behind the different period. The revisions provide a more physically grounded explanation of the different periods and their classification. Please see the revised manuscript (Section 3.3, Lines 383-495) for details.

**[Comment 28]** F1: are these enhancement measured all at the same wavelength?

**Response:** We appreciate the reviewer's insightful question. As different instruments were employed in these studies, the $E_{abs}$ shown in Figure 1 were not measured at exactly the same wavelength. However, to ensure consistency and minimize the influence of brown carbon (BrC) on the estimation of $E_{abs}$, we selected measurements obtained within the visible–near-infrared range for comparison. This wavelength selection greatly reduces the potential spectral bias associated with BrC absorption. The specific wavelengths used for each instrument have been summarized in Table S2.

**[Comment 29]** F2: please improve the labelling of the axis. Number counts and SD of…?

**Response:** We thank the reviewer for the helpful suggestion. The axis labels in Figure 2 have been revised to improve clarity. Specifically, "Number counts" has been replaced with "Normalized number counts of BC-containing particles", and "SD of …" has been updated to "SD of $\log_{10}(M_R)$".

**References:**

Bauer, S. E., Ault, A. P., and Prather, K. A.: Evaluation of aerosol mixing state classes in the GISS modelE-MATRIX climate model using single-particle mass spectrometry measurements, Journal of Geophysical Research: Atmospheres, 118, 9834-9844, 10.1002/jgrd.50700, 2013.

Bond, T. C. and Bergstrom, R. W.: Light Absorption by Carbonaceous Particles: An Investigative Review, Aerosol Science and Technology, 40, 27-67, 10.1080/02786820500421521, 2006.

Brooks, J., Liu, D., Allan, J. D., Williams, P. I., Haywood, J., Highwood, E. J., Kompalli, S. K., Babu, S. S., Satheesh, S. K., Turner, A. G., and Coe, H.: Black carbon physical and optical properties across northern India during pre-monsoon and monsoon seasons, Atmospheric Chemistry and Physics, 19, 13079-13096, 10.5194/acp-19-13079-2019., 2019.

Chen, G., Liu, C., Wang, J., Yin, Y., and Wang, Y.: Accounting for Black Carbon Mixing State, Nonsphericity, and Heterogeneity Effects in Its Optical Property Parameterization in a Climate Model, Journal of Geophysical Research: Atmospheres, 129, e2024JD041135, 10.1029/2024JD041135, 2024.

Fierce, L., Bond, T. C., Bauer, S. E., Mena, F., Riemer, N., and Univ. of Illinois at Urbana-Champaign, I. L.: Black carbon absorption at the global scale is affected by particle-scale diversity in composition, Nature Communications, 7, 12361-12361, 10.1038/ncomms12361, 2016.

Fierce, L., Onasch, T. B., Cappa, C. D., Mazzoleni, C., China, S., Bhandari, J., Davidovits, P., Al Fischer, D., Helgestad, T., Lambe, A. T., Sedlacek, A. J., Smith, G. D., Wolff, L., Brookhaven National Lab, U. N. Y., and Pacific Northwest National Lab, R. W. A.: Radiative absorption enhancements by black carbon controlled by particle-to-particle heterogeneity in composition, Proceedings of the National Academy of Sciences, 117, 5196-5203, 10.1073/pnas.1919723117, 2020.

Gao, R. S., Schwarz, J. P., Kelly, K. K., Fahey, D. W., Watts, L. A., Thompson, T. L., Spackman, J. R., Slowik, J. G., Cross, E. S., Han, J. H., Davidovits, P., Onasch, T. B., and Worsnop, D. R.: A Novel Method for Estimating Light-Scattering Properties of Soot Aerosols Using a Modified Single-Particle Soot Photometer, Aerosol Science and Technology, 41, 125-135, 10.1080/02786820601118398, 2007.

Gysel, M., Laborde, M., Olfert, J. S., Subramanian, R., and Gröhn, A. J.: Effective density of Aquadag and fullerene soot black carbon reference materials used for SP2 calibration, Atmospheric Measurement Techniques, 4, 2851-2858, 10.5194/amt-4-2851-2011, 2011.

Huang, X.-F., Peng, Y., Wei, J., Peng, J., Lin, X.-Y., Tang, M.-X., Cheng, Y., Men, Z., Fang, T., Zhang, J., He, L.-Y., Cao, L. M., Liu, C., Zhang, C., Mao, H., Seinfeld, J. H., and Wang, Y.: Microphysical complexity of black carbon particles restricts their warming potential, One Earth, 7, 10.1016/j.oneear.2023.12.004, 2024.

Laborde, M., Schnaiter, M., Linke, C., Saathoff, H., Naumann, K. H., Möhler, O., Berlenz, S., Wagner, U., Taylor, J. W., Liu, D., Flynn, M., Allan, J. D., Coe, H., Heimerl, K., Dahlkötter, F., Weinzierl, B., Wollny, A. G., Zanatta, M., Cozic, J., Laj, P., Hitzenberger, R., Schwarz, J. P., and Gysel, M.: Single Particle Soot Photometer intercomparison at the AIDA chamber, Atmospheric Measurement Techniques, 5, 3077-3097, 10.5194/amt-5-3077-2012, 2012.

Liu, D., Allan, J. D., Young, D. E., Coe, H., Beddows, D., Fleming, Z. L., Flynn, M. J., Gallagher, M. W., Harrison, R. M., Lee, J., Prevot, A. S. H., Taylor, J. W., Yin, J., Williams, P. I., and Zotter, P.: Size distribution, mixing state and source apportionment of black carbon aerosol in London during wintertime, Atmospheric Chemistry and Physics, 14, 10061-10084, 10.5194/acp-14-10061-2014, 2014.

Liu, D., Whitehead, J., Alfarra, M. R., Reyes-Villegas, E., Spracklen, D. V., Reddington, C. L., Kong, S., Williams, P. I., Ting, Y.-C., Haslett, S., Taylor, J. W., Flynn, M. J., Morgan, W. T., McFiggans, G., Coe, H., and Allan, J. D.: Black-carbon absorption enhancement in the atmosphere determined by particle mixing state, Nature Geoscience, 10, 184-188, 10.1038/ngeo2901, 2017.

Liu, H., Pan, X., Liu, D., Liu, X., Chen, X., Tian, Y., Sun, Y., Fu, P., and Wang, Z.: Mixing characteristics of refractory black carbon aerosols at an urban site in Beijing, Atmospheric Chemistry and Physics,

20, 5771-5785, 10.5194/acp-20-5771-2020, 2020.

Modini, R. L., Corbin, J. C., Brem, B. T., Irwin, M., Bertò, M., Pileci, R. E., Fetfatzis, P., Eleftheriadis, K., Henzing, B., Moerman, M. M., Liu, F., Müller, T., and Gysel-Beer, M.: Detailed characterization of the CAPS single-scattering albedo monitor (CAPS PMssa) as a field-deployable instrument for measuring aerosol light absorption with the extinction-minus-scattering method, Atmos. Meas. Tech., 14, 819-851, 10.5194/amt-14-819-2021, 2021.

Naseri, A., Corbin, J. C., and Olfert, J. S.: Comparison of the LEO and CPMA-SP2 techniques for black-carbon mixing-state measurements, Atmos. Meas. Tech., 17, 3719-3738, 10.5194/amt-17-3719-2024, 2024.

Qian, H., Xu, B., Xu, Z., Zou, Q., Zi, Q., Zuo, H., Zhang, F., Wei, J., Pei, X., Zhou, W., Jin, L., Tian, X., Zhao, W., and Wang, Z.: Anthropogenic Oxygenated Volatile Organic Compounds Dominate Atmospheric Oxidation Capacity and Ozone Production via Secondary Formation of Formaldehyde in the Urban Atmosphere, ACS ES&T Air, 2, 1033-1041, 10.1021/acsestair.4c00317, 2025.

Riemer, N., Ault, A. P., West, M., Craig, R. L., and Curtis, J. H.: Aerosol Mixing State: Measurements, Modeling, and Impacts, Reviews of Geophysics, 57, 187-249, 10.1029/2018RG000615, 2019.

Romshoo, B., Müller, T., Ahlawat, A., Wiedensohler, A., Haneef, M. V., Imran, M., Warsi, A. B., Mandariya, A. K., Habib, G., and Pöhlker, M. L.: Significant contribution of fractal morphology to aerosol light absorption in polluted environments dominated by black carbon (BC), npj Climate and Atmospheric Science, 7, 87, 10.1038/s41612-024-00634-0, 2024.

Schnaiter, M., Linke, C., Möhler, O., Naumann, K. H., Saathoff, H., Wagner, R., Schurath, U., and Wehner, B.: Absorption amplification of black carbon internally mixed with secondary organic aerosol, Journal of Geophysical Research: Atmospheres, 110, D19204-n/a, 10.1029/2005JD006046, 2005.

Stier, P., Feichter, J., Kinne, S., Kloster, S., Vignati, E., Wilson, J., Ganzeveld, L., Tegen, I., Werner, M., Balkanski, Y., Schulz, M., Boucher, O., Minikin, A., and Petzold, A.: The aerosol-climate model ECHAM5-HAM, Atmospheric Chemistry and Physics, 5, 1125-1156, 10.5194/acp-5-1125-2005, 2005.

Szopa, S., Naik, V., Adhikary, B., Artaxo, P., Berntsen, T., Collins, W. D., Fuzzi, S., Gallardo, L., Kiendler-Scharr, A., Klimont, Z., Liao, H., Unger, N., and Zanis, P.: Short-Lived Climate Forcers. In Climate Change 2021: The Physical Science Basis. Contribution of Working Group I to the Sixth Assessment Report of the Intergovernmental Panel on Climate Change (eds Masson-Delmotte, V. et al.), IPCC, 2021.

Wang, J., Wang, J., Cai, R., Liu, C., Jiang, J., Nie, W., Wang, J., Moteki, N., Zaveri, R. A., Huang, X., Ma, N., Chen, G., Wang, Z., Jin, Y., Cai, J., Zhang, Y., Chi, X., Holanda, B. A., Xing, J., Liu, T., Qi, X., Wang, Q., Pöhlker, C., Su, H., Cheng, Y., Wang, S., Hao, J., Andreae, M. O., and Ding, A.: Unified theoretical framework for black carbon mixing state allows greater accuracy of climate effect estimation, Nature Communications, 14, 2703, 10.1038/s41467-023-38330-x, 2023.

Wang, Y., Li, W., Huang, J., Liu, L., Pang, Y., He, C., Liu, F., Liu, D., Bi, L., Zhang, X., and Shi, Z.: Nonlinear Enhancement of Radiative Absorption by Black Carbon in Response to Particle Mixing Structure, Geophysical Research Letters, 48, 10.1029/2021gl096437, 2021.

Weber, P., Petzold, A., Bischof, O. F., Fischer, B., Berg, M., Freedman, A., Onasch, T. B., and Bundke, U.: Relative errors in derived multi-wavelength intensive aerosol optical properties using cavity attenuated phase shift single-scattering albedo monitors, a nephelometer, and tricolour absorption photometer measurements, Atmospheric Measurement Techniques, 15, 3279-3296, 10.5194/amt-

15-3279-2022, 2022.

Zanatta, M., Gysel, N., Bukowiecki, T., Müller, E., and Weingartner: A European aerosol phenomenology-5: Climatology of black carbon optical properties at 9 regional background sites across Europe, Atmospheric Environment, 2016.

Zanatta, M., Bogert, P., Ginot, P., Gong, Y., Hoshyaripour, G. A., Hu, Y., Jiang, F., Laj, P., Li, Y., Linke, C., Möhler, O., Saathoff, H., Schnaiter, M., Umo, N. S., Vogel, F., and Wagner, R.: AIDA Arctic transport experiment – Part 1: Simulation of northward transport and aging effect on fundamental black carbon properties, Aerosol Research, 3, 477-502, 10.5194/ar-3-477-2025, 2025.

Zeng, L., Tan, T., Zhao, G., Du, Z., Hu, S., Shang, D., and Hu, M.: Overestimation of black carbon light absorption due to mixing state heterogeneity, npj Climate and Atmospheric Science, 7, 2, 10.1038/s41612-023-00535-8, 2024.

Zhang, F., Shen, J., Xu, D., Shen, J., Qin, Y., Shi, R., Wei, J., Xu, Z., Pei, X., Tang, Q., Chen, H., Xu, B., and Wang, Z.: Unveiling the key drivers and formation pathways for secondary organic aerosols in an eastern China megacity, Journal of Hazardous Materials, 498, 10.1016/j.jhazmat.2025.139925, 2025a.

Zhang, Y., Zhang, Q., Cheng, Y., Su, H., Li, H., Li, M., Zhang, X., Ding, A., and He, K.: Amplification of light absorption of black carbon associated with air pollution, Atmospheric Chemistry and Physics, 18, 9879-9896, 10.5194/acp-18-9879-2018, 2018.

Zhang, Y., Zhang, Q., Cheng, Y., Su, H., Kecorius, S., Wang, Z., Wu, Z., Hu, M., Zhu, T., Wiedensohler, A., and He, K.: Measuring the morphology and density of internally mixed black carbon with SP2 and VTDMA: new insight into the absorption enhancement of black carbon in the atmosphere, Atmospheric Measurement Techniques, 9, 1833-1843, 10.5194/amt-9-1833-2016, 2016.

Zhang, Y. X., Zhang, Q., Yao, Z. L., and Li, H. Y.: Particle Size and Mixing State of Freshly Emitted Black Carbon from Different Combustion Sources in China, Environmental Science & Technology, 54, 7766-7774, 10.1021/acs.est.9b07373, 2020.

Zhang, Z., Wang, J., Wang, J., Riemer, N., Liu, C., Jin, Y., Tian, Z., Cai, J., Cheng, Y., Chen, G., Wang, B., Wang, S., and Ding, A.: Steady-state mixing state of black carbon aerosols from a particle-resolved model, Atmospheric Chemistry and Physics, 25, 1869-1881, 10.5194/acp-25-1869-2025, 2025b.

Zhao, G., Tan, T., Zhu, Y., Hu, M., and Zhao, C.: Method to quantify black carbon aerosol light absorption enhancement with a mixing state index, Atmos. Chem. Phys., 21, 18055-18063, 10.5194/acp-21-18055-2021, 2021.